# Quantifying the aerosol effect on droplet size distribution at cloud-top

Lianet  Hernández Pardo[1], Luiz Augusto Toledo Machado[1], Micael Amore Cecchini[1,2], and Madeleine Sánchez Gácita[1,3]

[1]Centro de Previsão de Tempo e Estudos Climáticos, Instituto Nacional de Pesquisas Espaciais, Cachoeira Paulista, Brasil
[2]Departamento de Ciências Atmosféricas, Instituto de Astronomia, Geofísica e Ciências Atmosféricas, Universidade de São Paulo, Brasil
[3]Centro de Ciência do Sistema Terrestre, Instituto Nacional de Pesquisas Espaciais, São josé dos Campos, Brasil

*Correspondence to:* Lianet Hernández Pardo (lianet.pardo@inpe.br)

**Abstract.** This work uses the number concentration-effective diameter phase-space to test cloud sensitivity to variations in the aerosol population characteristics, such as the aerosol size distribution, number concentration and hygroscopicity. It is based on the information from the top of a cloud simulated by a bin-microphysics single-column model, for initial conditions typical of the Amazon, using different assumptions regarding the entrainment and the aerosol size distribution. It is shown that the cloud-top evolution can be very sensitive to aerosol properties, but the relative importance of each parameter is variable. The sensitivity to each aerosol characteristic varies as a function of the tested parameter and is conditioned by the base values of the other parameters, showing an specific dependence for each configuration of the model. When both the entrainment and the bin treatment of the aerosol are allowed, the largest influence on the DSD sensitivity was obtained for the median radius of the aerosols and not for the total number concentration of aerosols. Our results reinforce that the CCN activity can not be predicted solely on the basis of the $w/N_a$ supersaturation-based regimes.

## 1   Introduction

Because of their role as cloud condensation nuclei (CCN) and ice nucleating particles, aerosols can affect the cloud optical properties (Twomey, 1974) and determine the onset of precipitation (Albrecht, 1989; Braga et al., 2017; Rosenfeld et al., 2008; Seifert and Beheng, 2006) and ice formation (Andreae et al., 2004; Fan et al., 2007; Gonçalves et al., 2015; Khain et al., 2005; Koren et al., 2010; Lee et al., 2008; Li et al., 2011). Aerosols also play an indirect role in the thermodynamics of local cloud fields through the suppression of cold pools and enhancement of atmospheric instability (Heiblum et al., 2016). However, knowledge about the characteristics of the effects of atmospheric aerosols on clouds and precipitation is still lacking and remains an important source of uncertainty in meteorological models.

Many studies have been dedicated to quantifying the effect of aerosols on clouds through sensitivity calculations, using both modeling and observational approaches. Knowing the real values of each parameter that characterize the aerosol is difficult. Also, detailed modeling of droplet nucleation implies a high computational cost. Thus, sensitivity studies intend to determine

whether the variability of some characteristics of the aerosol population can be neglected without introducing significant errors in the description of clouds.

A major debate refers to the relative importance of aerosol composition against size distribution and total number concentration (McFiggans et al., 2006). Several studies suggest that accurate measures of aerosol size and number concentration are more important to obtain a relatively accurate description of cloud droplet populations (Feingold, 2003; Dusek et al., 2006; Ervens et al., 2007; Gunthe et al., 2009; Rose et al., 2010; Reutter et al., 2009). However, other observations/simulations show that, under certain circumstances, neglecting the variability of the aerosol composition prevent realistic estimations of the aerosol effect on clouds (Hudson, 2007; Quinn et al., 2008; Cubison et al., 2008; Roesler and Penner, 2010; Sánchez Gácita et al., 2017). This circumstantial sensitivity is commonly found in the literature and it refers not only to aerosol composition, but also to other meteorological/aerosol conditions (McFiggans et al., 2006). For instance, Feingold (2003) showed that the influence of aerosol parameters over the droplet effective radius ($r_e$) varies as a function of aerosol loading. Under clean condition, $r_e$ is mostly determined by the liquid water content and the aerosol number concentration ($N_a$), with decreasing dependence on the aerosol size distribution (PSD), aerosol composition and vertical velocity ($w$). However, under polluted conditions, all of them contribute significantly to $r_e$. Reutter et al. (2009) obtained that the variability of the initial cloud droplet number concentration ($N_d$) in convective clouds is mostly dominated by the variability of $w$ and $N_a$. They found that the hygroscopicity parameter ($\kappa$) appears to play important roles at very low supersaturations in the updraft-limited regime of CCN activation. Also, a significant sensitivity of $N_d$ on the PSD parameters was found for all $w - N_a$ regimes under certain conditions. Karydis et al. (2012) used a global meteorological model to obtain the sensitivity field of $N_d$ to $w$, uptake coefficient, $\kappa$ and $N_a$. They state that, overall, $N_d$ is predicted to be less sensitive to changes in $\kappa$ than in $N_a$, although there are regions and times where they result in comparable sensitivities.

To further evidence the importance of aerosol composition on clouds, Ward et al. (2010) consider the Reutter et al. (2009) environmental regimes but vary the log-normal median aerosol radius ($\bar{r}_a$) to examine the behavior of the sensitivity to $\kappa$. Their results compare well with the Reutter et al. (2009) regime designation when using the same value of $\bar{r}_a$. However, they show that $w/N_a$, or supersaturation-based regimes, cannot fully predict the compositional dependence of CCN activity, it also varies significantly as a function of $\bar{r}_a$. It is remarkable that for small aerosols ($\bar{r}_a < 0.06\mu$m), composition affects CCN activity even in the aerosol-limited regime.

Previous researches investigating the aerosol effect on clouds have employed adiabatic parcel models to perform multiple sensitivity calculations (Feingold, 2003; Reutter et al., 2009; Ward et al., 2010). While that approach can capture the pure response of cloud-base DSDs to aerosols through droplet nucleation and activation, it lacks the representation of the complex interactions that govern the evolution of DSDs in real clouds. Allowing to represent turbulent mixing in the models can introduce significant departure from the results obtained under an adiabatic assumption. For instance, the entrained air is expected to decrease the buoyancy of the parcel through the transfer of both sensitive and latent heat, therefore reducing the updraft velocity. The consequent reduction of the supersaturation, as well as the increased availability of unactivated aerosols can enhance the water vapor competition in the cloud. Therefore, the responses of the system to changes in the aerosol properties can suffer notable variations when turbulence and mixing is considered.

Also, most of the previous studies are based on the information from cloud-base. However, given the possibility of occurrence of cloud-top nucleation (Sun et al., 2012), it would be useful to assess the evolution of the cloud-top droplet size distribution (DSD), along with the cloud-base DSD, for exploring the aerosol first indirect effect. In a growing cumulus, the cloud-top represents the beginning of the cloud development at each level, including cloud-base (because, in the initial stage of the cloud life-cycle, both the base and the top coincide in space). Thus, the characteristics of the DSD at cloud-top will strongly impact the evolution of the cloud, modulating the rates of microphysical process onward and therefore determining the structure of the cloud. As Cecchini et al. (2017) pointed out, studies should take into account the altitude above cloud base. The authors showed that, on average, droplet growth with cloud evolution is comparable in absolute value and is opposite to the aerosol effect. They determined that the aerosol effect on DSD shape inverts in sign with altitude, favoring broader droplet distributions close to cloud base but narrower DSDs higher in the clouds.

Another feature that is relatively common in cloud physics modelling studies is to treat the aerosol specie as a single-moment bulk variable, i.e. considering only one bin for the aerosol number concentration, that is log-normally distributed at each grid point and time step. Thus, the growth of wet aerosols is not resolved, and aerosols with dry sizes larger than the critical size defined by the Köhler equation are immediately added to the first bin of the DSD. By fixing the shape of the PSD, those models guarantee a continuous supply of larger aerosols for activation. Although the number concentration of aerosols decreases according to the amount of activated droplets, the assumed log-normal shape implies the continuous presence of particles in the right tail of the PSD. Also, by always assigning the activated droplets to the smallest bin of the DSD, a very narrow shape is induced, spending a longer time to grow by diffusion until the collision-coalescence rate increases.

With ample water vapor supply, high temperatures and a wide spectrum of aerosol conditions, the troposphere over the Amazon constitutes an ideal scenario to study aerosol-cloud-precipitation interaction. The Amazonian clouds that form during the wet and transition seasons are found to be very sensitive to aerosols (Andreae et al., 2004; Cecchini et al., 2016; Braga et al., 2017; Cecchini et al., 2017; Fan et al., 2018; Reid et al., 1999). Recent experimental campaigns in the Amazon have highlighted another layer of complexity in the aerosol-cloud interactions. During the wet season when the atmosphere is at the background aerosol conditions, the clouds control both the removal and production of atmospheric particles over the Amazon basin. According to Andreae et al. (2018), the production of new aerosol particles from biogenic volatile organic material, brought up by deep convection to the upper troposphere, is the dominant process supplying secondary aerosol particles in the pristine atmosphere. Then, those particles can be transported from the free troposphere into the boundary layer by strong convective downdrafts or even weaker downward motions in the trailing stratiform region of convective systems (Wang et al., 2016). During the transition or dry seasons, frequent biomass burning events change the aerosol population characteristics as a whole, not only their number concentrations. Therefore, it is important to infer the pollution effect on cloud properties and how they can interact with the natural cycle in the region.

Here we propose to explore the cloud sensitivities to several aerosol properties, by simulating some characteristics of Amazon clouds. We focus on the information from cloud-top, during the warm stages of cloud life-cycle, using a sample strategy that also includes the information from the cloud-base at the initial stage of development of the cloud. Our approach is similar to Ward et al. (2010), but it is not limited to analyze the hygroscopicity sensitivity. Instead, we extended the discussion to

the sensitivity to the aerosol median size and number concentration too, and consider their effects on both droplet size and concentration. This analysis is performed with three different model configurations that allow us to investigate the importance of representing the entrainment and mixing, as well as the evolution of the PSD, in modelling studies related to the aerosol effect.

## 2 Modelling approach

The simulations performed here employs variations of the Tel Aviv University (TAU) bin microphysics parameterization (Feingold et al., 1988; Tzivion et al., 1987, 1989) coupled to a single-column Eulerian framework. The 1D model is based on the Kinematic Driver (KiD) (Shipway and Hill, 2012), but instead of prescribing $w$ for each time $t$ and height $z$, it is calculated from the simplified vertical momentum equation, considering the buoyancy of the parcel and the weight of the liquid water, as well as the reaction force on the parcel resulting from the acceleration of the air in the neighborhood (Pruppacher and Klett, 2012):

$$\frac{dw}{dt} = \frac{g}{1+\gamma}\left(\frac{\theta - \theta'}{\theta'} - q_l\right) - \frac{\mu}{1+\gamma}w^2 \tag{1}$$

where $\gamma \equiv m'/2m \approx 0.5$, $m$ and $m'$ being the mass of the parcel and the mass of the air displaced by the parcel, respectively; $g$ is the gravity acceleration; $\theta$ and $\theta'$ are the potential temperature of the ascending parcel and the environment, respectively; and $q_l$ is the liquid water mixing ratio. The entrainment rate $\mu \equiv \frac{1}{m}\frac{dm}{dz}$ considers the lateral mass flux along the axis of a vertical plume of radius $R(t,z)$. It is assumed to follow the inverse radius dependence: $\mu = \frac{C}{R}$, where $C \approx 0.2$ is the entrainment parameter. The equation for the radius of the plume is:

$$\frac{d\ln R}{dt} = \frac{1}{2}\left(\mu w - \frac{d\ln\rho}{dt} - \frac{d\ln w}{dt}\right) \tag{2}$$

where $\rho$ represents the density of the air.

In our simulations, a 1s time step was used for both dynamics and microphysics algorithms during an integration time of 1800s (30min). For the vertical domain, a 120-level grid was defined with a 50-m grid spacing from 0m to 6000m of altitude.

As initial conditions, vertical profiles of potential temperature and water vapor mixing ratio ($q_v$) from an in situ atmospheric sounding corresponding to 1730Z on September 11, 2014, from Manacapuru, Brazil (Fig. 1) were provided. A constant temperature perturbation of 2.5K was introduced at surface to force the convection.

The contribution of the entrainment in the equations for the evolution of $\theta$, $q_v$ and $N_a$ is expressed as $\mu(X - X')w$, where $X$ and $X'$ represent the in-cloud and environment values for each one of the mentioned magnitudes, respectively.

### 2.1 Microphysics representation

For the simulations performed in this work, we have used the TAU size-bin-resolved microphysics scheme that was first developed by Tzivion et al. (1987, 1989) and Feingold et al. (1988) with later applications and development documented in

Stevens et al. (1996); Reisin et al. (1998); Yin et al. (2000a, b) and Rotach and Zardi (2007). TAU differs from other bin microphysical codes because it solves for two moments of the drop size distribution in each of the bins rather than solving the equations for the explicit size distribution at each mass/size point, which allows for a more accurate transfer of mass between bins and alleviates anomalous drop growth.

In this version of the TAU microphysics[1], the cloud drop size distribution is divided into 34 mass-doubling bins with radii ranging between $1.56\mu$m and $3200\mu$m. The method of moments (Tzivion et al., 1987) is used to compute mass and number concentrations in each size bin resulting from diffusional growth (Tzivion et al., 1989), collision-coalescence and collisional breakup (Tzivion et al., 1987; Feingold et al., 1988). Sedimentation is performed with a first-order upwind scheme.

To account for changes in the PSD, we introduced a set of 19 bins for dry aerosols, with radii ($r$) between 0.0076 and $7.6\mu$m, 
according to Kogan (1991). We consider that the total number concentration of aerosols is log-normally distributed through those bins, at the beginning of the simulation, and can vary by advection, entrainment, activation and regeneration after droplet evaporation.

At a given temperature and supersaturation, the critical dry size ($r_c$) for droplet activation is computed from the Köhler equation (Pruppacher and Klett, 2012). The initial bin for newly nucleated droplets is assigned according to its equilibrium 
size at 100% relative humidity, if $r < 0.09w^{-0.16}$. For larger aerosols, the initial radius of the droplet will exceed $r$ by a factor of $k = 5.8w^{-0.12}r^{-0.214}$, due to the time these particles take to reach its equilibrium size (Ivanova, 1977). The consumption of water vapor by unactivated aerosols is not considered in the model. We assume that aerosols smaller than the activation size do not represent a significant sink of water vapor, given the great availability of humidity over the Amazon.

The aerosol regeneration is included here following the approach of Kogan et al. (1995) and Hill et al. (2008). It considers 
that large CCN particles grow to large cloud drops, which evaporates less efficiently than small droplets. Thus, small CCN will be released before large ones. As a result, the regenerated CCN are replenished to the aerosol bins starting by the smallest activated size, until the original number concentration in each bin is attained. If the number concentration of regenerated CCN is larger than the number concentration of "missing" aerosols (considering the initial PSD), which can happen by advection of droplets to levels different than those where they were nucleated, the "excess" of CCN will be log-normally distributed 
according to the initially defined median radius and geometric standard deviation. A constraint is added to this scheme to conserve the domain-averaged PSD.

This scheme provides a reasonable way to parameterize the aerosol regeneration without using a two dimensional probability density function to track the aerosols. It does not consider the processing of the aerosols inside the cloud, therefore, it could induce errors in the activation rate in situations where the collision-coalescence process is a significant sink of small aerosols 
and a source of larger aerosols (Lebo and Seinfeld, 2011). However, its use is justified in our case because of the occurrence of only low rates of evaporation. This evaporation takes place right above cloud-top, due to the advection of droplets to upper, unsaturated levels. Hence, even if the collision-coalescence significantly modify the size of the aerosol particles, when partial evaporation occurs, only the smallest droplets will deactivate. The collision-coalescence effect on the PSD would have to

---

[1]Available at https://www.esrl.noaa.gov/csd/staff/graham.feingold/code/ (Accessed on: 04/11/2017)

be considered in cases with large evaporation rates, where even large droplets, containing the largest original or processed aerosols, deactivate.

## 3 Sensitivity analysis

We employ a phase space defined by two bulk properties of the DSD (hereinafter "bulk phase space"): $N_d$ (cm$^{-3}$), which coincides with the zeroth moment of the DSD, and $D_{eff}$ ($\mu$m), which is the ratio between the third and second moments.

Sensitivity tests in the bulk phase space provide a very efficient means to evaluate how a specific parameter variability can affect the evolution of cloud-top DSDs. Here, we test the sensitivity of $N_d$ and $D_{eff}$ at the cloud top to variations in $N_a$, $\bar{r}_a$, the geometric standard deviation ($\sigma_a$) of the PSD and $\kappa$, using ranges normally found in the Amazon atmosphere (Gunthe et al., 2009; Martin et al., 2010; Pöhlker et al., 2016) (Table 1). There are two sets of parameters tested. The set 1 applies to the tests employing bins for the aerosol, while the set 2 is used for the simulations with a bulk treatment of the aerosol.

The choice of the intervals of values for the aerosol properties was made in a way that allowed to explore the largest subset of realizable values of the parameters, while keeping a reasonable computation time. For certain combinations of the size distributions parameters, the PSD can be very narrow, with a very small concentration of aerosols larger than the activation threshold. This configuration, along with a small $N_a$, generates clouds with very low water content and unrealistically high supersaturations, when a bin treatment of the aerosol is used. To prevent this kind of situations, ranges were chosen as to produce averaged $N_d$ at cloud top >10cm$^{-3}$, while keeping the largest variety possible for each parameter. In order to test the response to $\bar{r}_a$=0.05$\mu$m, for instance, we needed to use values of $N_a$ larger than 800cm$^{-3}$, and $\sigma_a$ larger than 1.6. We did obtain realistic outputs from simulations with lower $N_a$, such as 200cm$^{-3}$, but only using PSDs with larger $\bar{r}_a$ and $\sigma_a$. Since we needed a fully applicable parameter-space, it explains the choice of the described intervals.

Due to a deficient treatment of the activation scavenging, when a bulk treatment of the aerosol is used, the lower values of the aerosol parameters at which a reasonably dense cloud can be generated are much smaller. By not allowing the PSD to freely evolve, there is a continuous, spurius source of large aerosols that induces unrealistically high values of $N_d$ and can unstabilize the model if some thresholds for $N_a$, $\bar{r}_a$ and $\sigma_a$ (900cm$^{-3}$, 0.08$\mu$m and 1.9, respectively) are exceeded. Therefore, in this case, the upper and lower limits for each parameter had to be decreased.

The sensitivities were calculated as the slope of the linear fit between $Y$ and $X_i$ in logarithmic scale for normalization:

$$S_Y(X_i) = \frac{\partial \ln Y}{\partial \ln X_i}\bigg|_{X_k} \tag{3}$$

where $Y$ represents either $N_d$ or $D_{eff}$, and $X_i$ is the aerosol property affecting $Y$. $S_Y(X_i)$ represents the relative change in $Y$ for a relative change in $X_i$ and places less reliance on the absolute measures of parameters (Feingold, 2003; Reutter et al., 2009; Ward et al., 2010). The subscript $X_k$ indicates that when calculating the sensitivity to $X_i$, the other aerosol parameters are held constant. For each value at which $X_k$ is fixed, we will obtain a new value of $S_Y(X_i)$, i.e. we can also calculate $S_Y(X_i)$ as a function of $X_k$ ($S_Y(X_i, X_k)$).

**Table 1.** Aerosol parameters used for the sensitivity tests using bin and bulk approaches for the aerosol: intervals for values and steps between them. For additional details, the reader is referred to the text.

| Parameter | Set 1: bin | | Set 2: bulk | |
| --- | --- | --- | --- | --- |
| | Interval | Step | Interval | Step |
| $N_a$ (cm$^{-3}$) | $800 - 3600$ | 400 | $200 - 900$ | 100 |
| $\bar{r}_a$ ($\mu$m) | $0.05 - 0.11$ | 0.01 | $0.02 - 0.08$ | 0.01 |
| $\sigma_a$ () | $1.6 - 2.2$ | 0.1 | $1.1 - 1.9$ | 0.2 |
| $\kappa$ () | $0.1 - 0.5$ | 0.1 | $0.1 - 0.5$ | 0.1 |

The latter differentiates our approach from previous studies. Feingold (2003) included the variability of all $X \neq X_i$ when calculating the linear regression between $\ln Y$ and $\ln X_i$, only distinguishing the results for two subsets of $N_a$. Similarly, Reutter et al. (2009) analyzed the sensitivities to $\bar{r}_a$, $\sigma_a$ and $\kappa$ for three combinations of $N_a$ and $w$, but all values of Y calculated at a given value of $X_i$ were averaged prior to fitting. This analysis was then expanded by Ward et al. (2010), who calculated $S_{N_d}(\kappa)$
for different values of $\bar{r}_a$ and $\sigma_a$ used to initialize the parcel model. Now, we use a more general approach that allows us to study the responses of both cloud droplet number concentration and effective diameter to changes in each aerosol characteristic, as a function of the other aerosol parameters used to initialize the model.

## 4 Results

The control run of the model produced a shallow cumulus that grew to 4000 m depth in about 30 minutes. Fig. 2 shows the
evolution of the updrafts, droplet concentration and effective diameter, characterized by the following aerosol initial parameters: $N_a$=800cm$^{-3}$, $\bar{r}_a$=0.08$\mu$m, $\sigma_a$=1.9, and $\kappa$=0.1.The cloud-top is defined as the last model level, from surface to top, where the droplet concentration was larger than 1 per cm$^3$. Note that there is a maximum of $N_d$ at cloud-top for all times. As droplets ascend and mix with new droplets, they grow by diffusion of vapor and collision-coalescence. As a consequence, $D_{eff}$ is larger in upper levels.
The bulk phase space view is introduced in Fig. 3 to discuss the isolated effect of each parameter, when keeping the other aerosol PSD properties constant. Overall, following the cloud-top in the phase-space, two local maximums of $N_d$ are found. The first one corresponds to the smallest $D_{eff}$ ($< 5\ \mu$m) and is related to the maximum in the nucleation rate. This represents the first steps in cloud formation, where the droplets are very small and there is no significant vertical cloud development. The second one, which is also the global maximum, is reached when the cloud is deeper, as a consequence of the accumulation of
droplets advected by the updraft. Regardless of the $N_d$ fluctuations, the cloud-top $D_{eff}$ shows an overall monotonic increase with altitude, except in the end of the simulation where the updraft decelerates.

Figure 3a shows the sensitivity of cloud-top DSDs to the initial concentration of aerosols. Note that an increase of $N_a$ increases $N_d$ for the most part, as expected. The nucleation enhancement induces a smaller $D_{eff}$ because of water vapor

competition, for the same liquid water content (not shown). Thus, if the water vapor amount is kept constant, the diffusional growth for each droplet is slowed. The latter manifests as a trend to the horizontal orientation in the lower portion of the trajectories in the bulk phase space, corresponding to the smallest sizes ($< 10\ \mu$m), where diffusion of water vapor is the predominant droplet growth mechanism. It is interesting to note that all profiles evolve towards similar values in their maximum

$N_d$ and $D_{eff}$. This is related to a buffering effect of the entrainment. Note that the entrainment term, in the temperature, water vapor and aerosol tendency equations, is proportional to the difference of the values of those variables between the cloud and the environment. Larger aerosol content will induce strongest modifications in the fields, thus increasing the contribution of the entrainment term. This feedback effect decreases the sensitivity of the maximum $N_d$ and $D_{eff}$ attained in the cloud to the aerosol loading. Also notable are the $N_d > N_a$ values in the control run, which results from the vertical gradient of $w$ shown

in Fig. 2. Because the updrafts are stronger below cloud top, there is a tendency to accumulate droplets in the layers analyzed here.

Note that the fraction of activated droplets in the first level is similar between all simulations in Fig. 3a (close to one third of $N_a$), which is a reflect of all other aerosol PSD parameters being kept constant. In reality, increased pollution in the Amazon is usually followed by changes in aerosol PSD shape, given the different properties of background and biomass-burning or urban

particles. Therefore, it is important to analyze the effects of every aerosol PSD parameter separately to fully understand the pollution effect in Amazonian clouds.

Figure 3b and 3c show the sensitivity of cloud-top DSDs to $\bar{r}_a$ and $\sigma_a$ while keeping the other parameters at their control standards. The effects of increasing aerosol size and PSD width are similar to the consequences of increasing $N_a$. By increasing $\bar{r}_a$ or $\sigma_a$, more droplets are activated because of the larger availability of aerosols with sizes above the activation threshold.

Thus, nucleation increases, whereas diffusional growth decreases. The latter is visible during the entire trajectories in Fig. 3b and 3c.

The tests in Fig. 3 evidence a type of saturation effect for the larger values of $N_a$, $\bar{r}_a$ and $\sigma_a$ tested, i.e. the sensitivity decreases as these parameters increase. This behavior is explained mainly by the supersaturation consumption. Even if continuous water vapor supply from the surface occurs, the supersaturation can be completely consumed, depending on the aerosol

availability and the diffusional growth rate. If the number of activated aerosols is able to consume all the supersaturation, given certain $z$ and $t$, an increase of its quantity will not introduce differences in the DSD.

Finally, Fig. 3d shows that the effects of varying $\kappa$ are very small. Nevertheless, this is a result for one single combination of $N_a$, $\bar{r}_a$ and $\sigma_a$, i.e., the control values of the parameters; according to Ward et al. (2010), the sensitivity to $\kappa$ can vary as a function of $N_a$ and $\bar{r}_a$. Additionally, it is known that the sensitivity to $\kappa$ increases substantially as $\kappa$ decreases (Petters and

Kreidenweis, 2007). However, that effect is more or less evident depending on the values of the other parameters. Hence, to characterize the sensitivity of DSDs to aerosol properties, we should explore the multiparameter space composed by all combinations of discrete values of the parameters from its interval of realizable values.

To illustrate that sensitivity variation, we calculated $S_{\bar{N}_d}(X_i)$ and $S_{\bar{D}_{eff}}(X_i)$, with $X_i$ being $N_a$, $\bar{r}_a$, $\sigma_a$ or $\kappa$. $\bar{N}_d$ and $\bar{D}_{eff}$ are the time averages of $N_d$ and $D_{eff}$ at cloud-top for each simulation, respectively. From Eq. 3, $S_{\bar{N}_d}(N_a)$, for example, is

the slope of the linear fit between the values of $\bar{N}_d$ and $N_a$ in logarithmic scale, for a given combination of $\bar{r}_a$, $\sigma_a$ and $\kappa$. The

sensitivity to one aerosol parameter can then be calculated a number of times equivalent to all possible combinations of the values of the other parameters in Table 1.

Figures 4, 5, 6 and 7 show $S_Y(X_i)$ as a function of all values of $N_a$, $\bar{r}_a$, $\sigma_a$ and $\kappa$ considered. Generally, $\bar{N}_d$ can be almost three times more sensitive to changes in the aerosol parameters than $\bar{D}_{eff}$, which stems from the mathematical definition of these physical magnitudes. For each value in the x-axis of figures 4, 5, 6 and 7, there are several combinations of the other two parameters; as a result, there are several points for each value of the x-axis in the figures.

The impact of $N_a$ on cloud droplets depends on the values of $\bar{r}_a$ and $\sigma_a$, but does not vary with $\kappa$, as can be seen in Fig. 4. For smaller values of $\bar{r}_a$ and $\sigma_a$, $S_Y(N_a)$ reaches its maximum and presents a large dispersion. On the other hand, it tends to be concentrated around a minimum sensitivity value as these parameters increase. Hence, for smaller aerosols, the relative importance of the aerosol properties can be very different to that at larger sizes.

Figure 5a shows that the sensitivity to the median radius of the aerosol population decreases for higher values of $N_a$ and $\sigma_a$. Similar to the behavior of $S_Y(N_a)$, the lower variability in $S_Y(\bar{r}_a)$ corresponds to the values of $N_a$ and $\sigma_a$ where the absolute value of the mean sensitivity is minimum. Conversely, the effects of $\kappa$ on the sensitivity to $\bar{r}_a$ are negligible (Fig. 5c).

The same applies to the sensitivity to $\sigma_a$ (Fig. 6), substituting $\sigma_a$ by $\bar{r}_a$ as the independent variable in Fig. 6b. It is remarkable that the absolute values of $S_Y(\sigma_a)$ are the highest between those analyzed here. Nevertheless, even when $S_Y(\sigma_a)$ indicates having a high relative impact on $\bar{N}_d$ and $\bar{D}_{eff}$ for certain circumstances, we should keep in mind that the effect of varying a parameter is determined by its range of realizable values. For example, assuming that the maximum and minimum values specified in Table 1 determine the entire variation of the parameters in a given situation, it follows that a 0.6 change in $\sigma_a$ (an increase ratio of 1.38) could induce a 10.6 times increase in $\bar{N}_d$, while a variation of $0.06\mu m$ in $\bar{r}_a$ (a 2.2 increase ratio) can increase $\bar{N}_d$ 21.6 times, if we consider the maximum values of $S_{\bar{N}_d}(\sigma_a)$ and $S_{\bar{N}_d}(\bar{r}_a)$, respectively. In turn, a $2800\mathrm{cm}^{-3}$ change in $N_a$ (corresponding to a 4.5 increase ratio), would only increase $\bar{N}_d$ by a factor of 6.1 at most.

Note that $S_Y(\sigma_a)$ changes its sign as $\bar{r}_a$ increases (Fig. 6b). This is related to variations in the effect of $\sigma_a$ depending on the relation $\frac{r_c}{\bar{r}_a}$. Considering a log-normal PSD, the number of aerosols for which $r > r_c$, i.e., the number of activated droplets, is positively correlated to $\sigma_a$ if $\frac{r_c}{\bar{r}_a} > 1$, and negatively correlated otherwise. If $\frac{r_c}{\bar{r}_a} = 1$, the number of activated droplets does not depend on $\sigma_a$. The positive values obtained by Feingold (2003) for the sensitivity of droplet size on $\sigma_a$, as well as the negative values reported by Reutter et al. (2009) for the sensitivity of droplet number concentration on $\sigma_a$ should be due to the inclusion of larger aerosols, favoring the diminution of the $r_c$-to-$\bar{r}_a$ ratio.

Finally, the sensitivity to the aerosol hygroscopicity is the lowest between those analyzed here (Fig. 7). Note that an increase ratio of 5 in the value of $\kappa$ modifies $\bar{N}_d$ by a factor of 1.38 at most. This is also consistent with its small influence on the sensitivities of the other parameters, as mentioned above. The symmetric distribution of the sensitivity with respect to the abscissas axis evidences a random impact of $\kappa$ on the cloud-top DSDs here. This randomness is a reflect of the uncertainties involved in the determination of the cloud-top location, the calculation of $\bar{N}_d$ and $\bar{D}_{eff}$, as well as in the fitting procedure employed to obtain $S_Y(\kappa)$, that predominate in the presence of such low values of $S_Y(\kappa)$. However, it should be considered that the effects of the aerosol composition can be significantly increased in conditions of weak updrafts (Ervens et al., 2005; Anttila and Kerminen, 2007; Reutter et al., 2009).

## 5   Discussion

Despite the limited dynamical capabilities of our 1D framework, we adopted here a simplified approach to consider the mixing between the in-cloud and environment properties. We consider that the column in the model is located in the center of a plume with radius $R(t, z)$, which mixes homogeneously with the radially entrained air at each $z$. The entrainment affects the vertical velocity, the temperature, the humidity and the amount of aerosols in the column. Past studies in the Amazon have assumed that the entrainment mixing in Amazonian clouds is close to the extreme inhomogeneous case, given that the droplet effective radius remain relatively constant horizontally (Freud et al., 2011). However, the recent studies of Pinsky et al. (2016); Pinsky and Khain (2018) indicate that homogeneous and inhomogeneous mixing can be indistinguishable for polydisperse DSDs, especially for wide distributions. Additionally, those studies show the inadequacy of previous in-situ techniques to identify mixing type (the so-called mixing diagrams). Based on this finding, we will stick to the homogeneous case in the present study as a first approximation. Further studies would be needed to assess the effects of inhomogeneous mixing and this comparison is beyond the scope of this manuscript.

Some cloud-top mixing is resolved in the model grid. However, it can be affected by the numerical diffusion and dispersion introduced by the scheme that solves the advective terms. The representativeness of the mixing induced by such an advection at cloud top must be analyzed carefully, and is out of the scope of this paper. For now, we limit our analysis to the results with and without the inclusion of some lateral entrainment rates, as a proxy for the effect of the dilution caused by mixing with the air in the neighbourhood of the clouds. By using bins for the aerosol, we allow the PSD to evolve freely. This way, after activation, the tail of the PSD can only be filled again if new particles are advected, entrained or replenished due to droplet evaporation. Also, since the newly activated droplets fill several bins of the DSD, the development of wider DSDs is favored, accelerating collection processes. This method has been extensively employed (Yin et al., 2000b, a, 2005; Altaratz et al., 2008; Hill et al., 2008; Mechem and Kogan, 2008) to substitute the explicit calculation of the diffusional growth of the aerosols from its dry sizes, which has a much higher computational demand. Leroy et al. (2007) analyzed the influence of a similar assumption on the liquid and ice water content and the aerosol particles, drops and ice crystal spectra simulated by a 1.5D model. He found notable consistency between both approaches, even when the bin resolution was strongly decreased, as well as a reasonable sensitivity to the initial aerosol spectra. We use this approach here to test the importance of including a more detailed treatment of the PSD in the model, when investigating the aerosol effect on cloud-top DSDs.

Figure 8 and 9 illustrate the behavior of the sensitivity to each aerosol parameter in three different hypothetical situations. The first column shows the results from the simulations described in the previous section, the results without entraiment are shown in the second column, and the simulations using a bulk approach for the aerosol (with entrainment) are represented in the third column. For the plots shown in the first three lines in Figs. 8 and 9, the value of $\kappa$ is fixed to 0.1. The response of the sensitivities to changes in $\kappa$ are not shown because of its smaller influence compared to the other parameters. The graphs in the last line in Figs. 8 and 9 show $S_Y(\kappa)$ at $\sigma_a = 1.9$ and $\sigma_a = 1.5$, for the cases with a bin and bulk treatment of the aerosol, respectively. The variations of $S_Y(\kappa)$ due to changes in $\sigma_a$ are similar to the variations due to $\bar{r}_a$, which is represented in the y-axis of the figures.

The values of the aerosol parameters in the tests without bins for the aerosols (third column in Figs. 8 and 9) are usually lower than in the previously discussed tests. The reason is that, with this configuration, when the original values of the parameters are used, there is a very high nucleation rate that leads to unrealistic values of $N_d$ and ends up by destabilizing the model. It is reasonable, considering that once the aerosol is removed from activation, the remaining unactivated aerosols are spread over all

sizes, perpetuating the conditions for droplet formation. At the same time, this permits clouds to develop in conditions where there would be a negligible nucleation rate if a bin treatment of the aerosol were employed.

Figure 8b,e,h,k show that, without entrainment, $S_{\bar{N}_d}(X_i)$ is lower for low values of $N_a$, $\bar{r}_a$ and $\sigma_a$, due to a faster depletion of the aerosols of suitable sizes for activation. A secondary decrease in the sensitivity is found at more polluted situations, with larger aerosols and wider sizes distributions. The latter effect is caused by the supersaturation depletion related to an

increase in the amount of activating aerosols. That behavior contrasts with the responses in the entrainment case, where the lower supersaturations and the supply of additional aerosols from the environment enhance the water vapour depletion and inhibits the aerosol depletion effects.

When the entrainment is not considered, $S_{\bar{D}_{eff}}(N_a)$ reaches very low absolute values or even positive values for an intermediate interval of the independent variables, and increases its absolute value otherwise (Fig. 9b). The same behavior is shown

for $S_{\bar{D}_{eff}}(\bar{r}_a)$ and $S_{\bar{D}_{eff}}(\sigma_a)$ (Figs. 9e and 9h). The positive sensitivity evidences a less intense water vapour competition. At those points, increasing the $N_a$, $\bar{r}_a$ and/or $\sigma_a$ will create more droplets, given the positive values of $S_{\bar{N}_d}(N_a)$, $S_{\bar{N}_d}(\bar{r}_a)$ and $S_{\bar{N}_d}(\sigma_a)$ discussed above, increasing the vertical velocity by latent heat release, and therefore the supersaturation. Thus, if the increment in the number of droplets is not as intense as needed to cause a significant water vapour depletion, all the droplets will grow in the presence of such high supersaturations, therefore increasing $D_{eff}$. Conversely, for the smallest values of $N_a$, $\bar{r}_a$

and $\sigma_a$, the sensitivity decreases its absolute value again or even becomes negative. In that situation, only the largest aerosols in the right tail of the PSD are activated. Larger drops have a slower rate of growth by condensation, and the collision-coalescence rate may also be decreased due to less variety of fall speeds. Thus, even at high supersaturations, the growth of these droplets can be slower. In addition, when the total number concentration is increased and the shape of the distribution is maintained, the largest increments in the amount of aerosol occur near the center of the size distribution (mode values). Now, let's consider

what happens in the right tail of the PSD, i.e., the aerosols that will be activated. In that situation, since the largest increments in number concentration occur toward the center of the distribution, the smaller sizes in the right tail will be favored, leading to a decrease in $D_{eff}$ after activation. If the droplets growth rate is not as intense as to balance that trend, it will result in negative sensitivity.

In Figs. 8c and 9c it can be seen that, when a bulk approach is used for aerosols, the absolute value of $S_Y(N_a)$ increases

monotonically as $\bar{r}_a$ and $\sigma_a$ increase and it is not affected by the supersaturation depletion, because independently of $r_c$, there will always be a certain amount of aerosols such that $r > r_c$.

Also, the absolute value of $S_Y(\bar{r}_a)$ increases for higher values of $N_a$, which coincides with the results of Feingold (2003) and Rissman et al. (2004), and for lower values of $\sigma_a$ (Figs. 8f and 9f). The same applies to $S_Y(\sigma_a)$ (Figs. 8i and 9i), substituting $\sigma_a$ by $\bar{r}_a$ as the independent variable. However, the maximum value of the sensitivity to the size-related parameters is significantly

decreased compared to the simulations with a bin treatment of the aerosol. $S_{\bar{N}_d}(\sigma_a)$ even reaches slightly negatives values

for the larger $\bar{r}_a$ in these tests, which is related to the previously commented variations in the effect of $\sigma_a$ depending on the position of $r_c$ with respect to the size distribution function.

Finally, it can be observed in Fig. 8l and 9l that, when the model uses a bulk approach for the aerosol specie, $S_Y(\kappa)$ is larger for higher $N_a$ and smaller $\bar{r}_a$, in agreement with the results of Ward et al. (2010). The figures evidence that $\bar{N}_d$ and $\bar{D}_{eff}$ are much more sensitive to $\kappa$ when considering a bulk approach for the aerosols than when its size distribution is explicitly represented in the model. Note that, in the former case, $S_Y(\kappa)$ can be about 50% of $S_Y(N_a)$, which is a significant influence. However, perhaps the most relevant difference between these simulations and the ones using bins for the aerosol is the change in the sign of $S_Y(\kappa)$. Although, at first, higher values of $\kappa$ would determine a smaller $r_c$, it also contributes to a faster depletion of the larger aerosols, leading to a reduction in the nucleation rate afterward. That is the cause for the negative (positive) values of $S_{\bar{N}_d}(\kappa)$ ($S_{\bar{D}_{eff}}(\kappa)$) obtained in the tests using bins for the aerosol (Fig. 8k and 9k). On the other hand, the latter has not effect on the results when the PSD is fixed, therefore positive (negative) values of $S_{\bar{N}_d}(\kappa)$ ($S_{\bar{D}_{eff}}(\kappa)$) are obtained.

Overall, our analysis shows that increases in $N_a$, $\bar{r}_a$ and $\sigma_a$ produce higher $\bar{N}_d$ (positive sensitivity) and smaller $\bar{D}_{eff}$ (negative sensitivity) when both entrainment and aerosol bins are included in the simulations. This coincides with the results of Cecchini et al. (2017), who found cloud-top averages of $S_{\bar{N}_d}(N_a)$ and $S_{\bar{D}_{eff}}(N_a)$ of 0.84 and -0.25, respectively, from aircraft measurements over the Amazon forest.

The values of sensitivities reported by Feingold (2003); Reutter et al. (2009); Ward et al. (2010) are included in the range of sensitivities obtained here; plus the variability added by the diverse universe of situations found over the aerosol parameter space. However, comparisons between our results and previous researches are not straigforward, considering the influence of the cloud evolution here. How fast is the aerosol and water vapor depletion, for example, will determine how much nucleation will occur above cloud-base. A large supersaturation can cause a fast activation rate initially, but will decrease the intensity of that process afterwards. The response to changes in the aerosol properties in this case might be different from that with a moderate and more spatially distributed activation rate. In the simulations with a bulk treatment of the aerosol, the aerosol depletion is slower. Thus, for a certain time interval, each cloud-top level behaves like an independent cloud base regarding the intensity of the nucleation. That explains the similarities between the sensitivities obtained from the simulations using a bulk approach for the aerosols and previous researches.

From our analysis, it turns out that $\bar{r}_a$ is the most influential parameter that determines the sensitivity to aerosols at cloud-top, in contrast with the importance that has been conventionally attributed to the aerosol number concentration. To further illustrate this, Figure 10 shows the mean and standard deviation of $\bar{N}_d$ and $\bar{D}_{eff}$ for each value of $N_a$ tested, at each of the above referenced situations: with entrainment and bin for aerosols (a-b), without entrainment (c-d), and without bins for aerosols (e-f). The length of the standard deviation bars reflects the changes in $\bar{r}_a$, $\sigma_a$ and $\kappa$.

For the first (and most complete) situation considered, it can be seen that the state of the system is not sufficiently determined by $N_a$, specially if the PSD is displaced to smaller radius (Fig. 10b). For instance, increasing $N_a$ by a factor of 3 in Fig. 10b, from $800\,\mathrm{cm}^{-3}$ to $2400\,\mathrm{cm}^{-3}$, there is still some overlapping between the corresponding standard deviation bars in the phase-space. However, the bars are significantly smaller if larger aerosols are considered (Fig. 10a), indicating a tendency to approach the generally accepted knowledge, i.e., increasing the importance of $N_a$ in determining the characteristics of the DSDs. These

results highlight the importance of including the aerosol size distribution characteristics in aerosol-cloud interaction studies, especially when $\bar{r}_a \leq 0.08 \ \mu$m. These parameters can produce changes in the DSD as large as those caused by changes in the aerosol number concentration. These findings are also relevant given the current discussion about the importance of ultrafine aerosol particles in the development of deep convective clouds over the Amazon (Wang et al., 2016; Fan et al., 2018).

In turn, Fig. 10c-d show that, when the entrainment is not considered in the model, the variability of $\bar{N}_d$ and $\bar{D}_{eff}$ does not present a significant dependence on the aerosol size; it is a function of $\bar{N}_d$ and $\bar{D}_{eff}$ on their own. In other words, the location of the points in the phase-space determines their standard deviation. Points located at the left upper corner in Fig. 10c, for instance, have approximately the same standard deviation than points at the same location in Fig. 10d. The difference between both graphics resides on the position of the points: for smaller aerosols (Fig. 10d), $\bar{N}_d$ will be lower and $\bar{D}_{eff}$ will be larger,

than for large aerosols (Fig. 10c). On the other hand, the results indicate that the importance of $N_a$ may be overestimated if a bulk treatment of the aerosol is employed (Fig. 10e-f). It can be seen that, in this case, there is a reduction of the overlapping between the standard deviation bars, specially for larger and sparser aerosols.

     The simulations performed here represent an idealized cloud resulting from observed humidity and temperature profiles. However, even if we assume it represent a realizable situation, corresponding to an average behavior, it does not include

the variety of possibilities existing in real cases. Important processes such as turbulent entrainment and dynamic feedbacks can introduce a significant departure from the idealization we are considering. Full dynamical models account for dynamics feedbacks and several subgrid processes that could enhance or reduce the range of sensitivities that are demonstrated here. Nevertheless, the qualitative behavior of our main results, i.e., the dependency of the DSD sensitivity to the aerosol properties according to its position in the full parameter space, might not change. For example, Gettelman (2015) simulated several warm

rain cases with the KiD model and climatological cases with a global model, using a double-moment microphysics scheme, in order to analyze the sensitivity of the aerosol-cloud interaction to cloud microphysics. They found that the test in the KiD were consistent with the global sensitivity tests. This is an aspect we intend to study in a following work, to build on the present results.

## 6   Summary and conclusions

We illustrated the influence of the aerosol number concentration, the median radius and geometric standard deviation of the PSD, and the hygroscopicity of the aerosols on the number concentration and effective diameter of droplets at the top of warm-phase clouds, for initial conditions typical of the Amazon. The sensitivities behaved accordingly with the relation between the supersaturaration and the aerosol availability, that determine the rate of aerosol activation, as described by Reutter et al. (2009). Nevertheless, in our analysis, the intensity of the droplet activation is mostly determined by the amount of suitable-sized

aerosols, i.e. the shape and median radius of the PSD, rather than on the total number concentration of aerosols.

     We showed that the sensitivity to each aerosol characteristic varies as a function of the tested parameter and its value depends on the base value of the other parameters. The median radius of the aerosols is the most important parameter, from those analyzed, that influences the sensitivity to the others. This expands the result of Ward et al. (2010) and states that $w/N_a$,

or supersaturation-based regimes (Reutter et al., 2009), cannot fully predict the dependence of CCN activity, not only on the aerosol composition, but on all aerosol characteristics.

Given the tested variations in the aerosol properties, the responses of the DSDs depend on the model assumptions regarding the entrainment and the treatment of the aerosol size distribution. This reinforces the importance of carefully considering the characteristics of the model when analyzing the responses to changes in aerosol loading in global or regional studies.

Overall, when the nucleation is favored, an increase in the droplet number concentration is accompanied by a decrease in the droplet effective diameter. However, since our sensitivity analysis involves the evolution of the cloud-top with time and height, the results are not directly comparable with previously reported sensitivity calculations at cloud-base. When a series of consecutive nucleation events is considered, such as those during the evolution of the cloud top, the intensity of the nucleation at certain time can modulate its intensity afterwards. The simulation with a bulk treatment of the aerosols constitutes an extreme case of slow aerosol depletion, where the responses of the nucleation to changes in the aerosol properties can impact the cloud-top in a more homogeneous way. That is the reason for the agreement in the sensitivity obtained from those simulations and previous cloud-base sensitivity calculations.

*Author contributions.* LHP performed the model simulations, the model–data analysis and prepared the manuscript. LATM and MAC provided guidance with the definition of the model initial conditions. LATM, MAC and MSG provided guidance with the choice of the variables and its interval of values and the model–data analysis. All authors contributed to the design of the study and the preparation of the manuscript.

*Competing interests.* The authors declare that they have no conflict of interest.

*Acknowledgements.* This research was funded by the SOS CHUVA FAPESP Project 2015/14497-0. The contributions of Micael A. Cecchini and Lianet H. Pardo were funded by FAPESP grants 2017/04654-6 and 2016/24562-6, respectively.

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

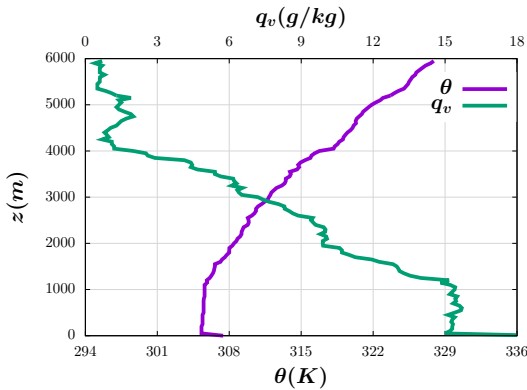

**Figure 1.** Vertical profiles employed as initial conditions in the simulations

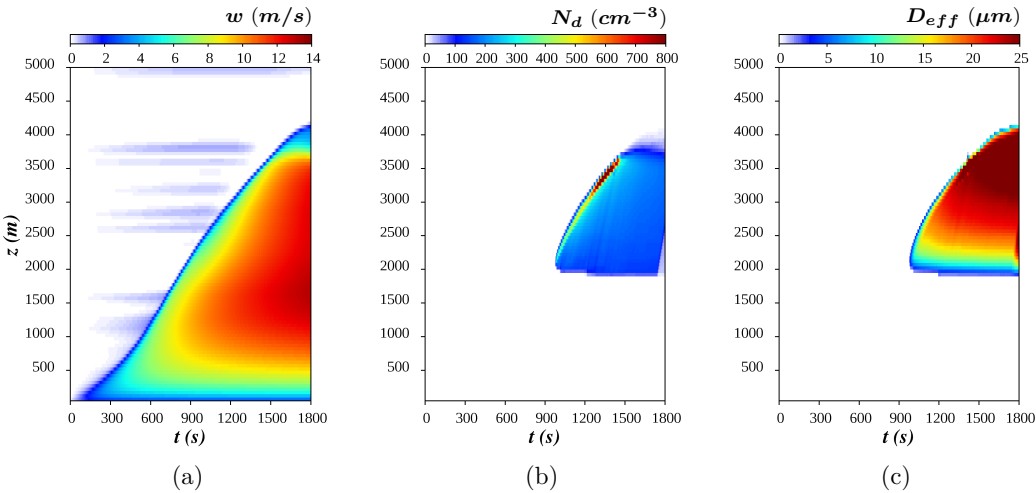

**Figure 2.** Evolution of $w$ (m/s), $N_d$ (cm$^{-3}$) and $D_{eff}$ ($\mu$m) in the simulation.

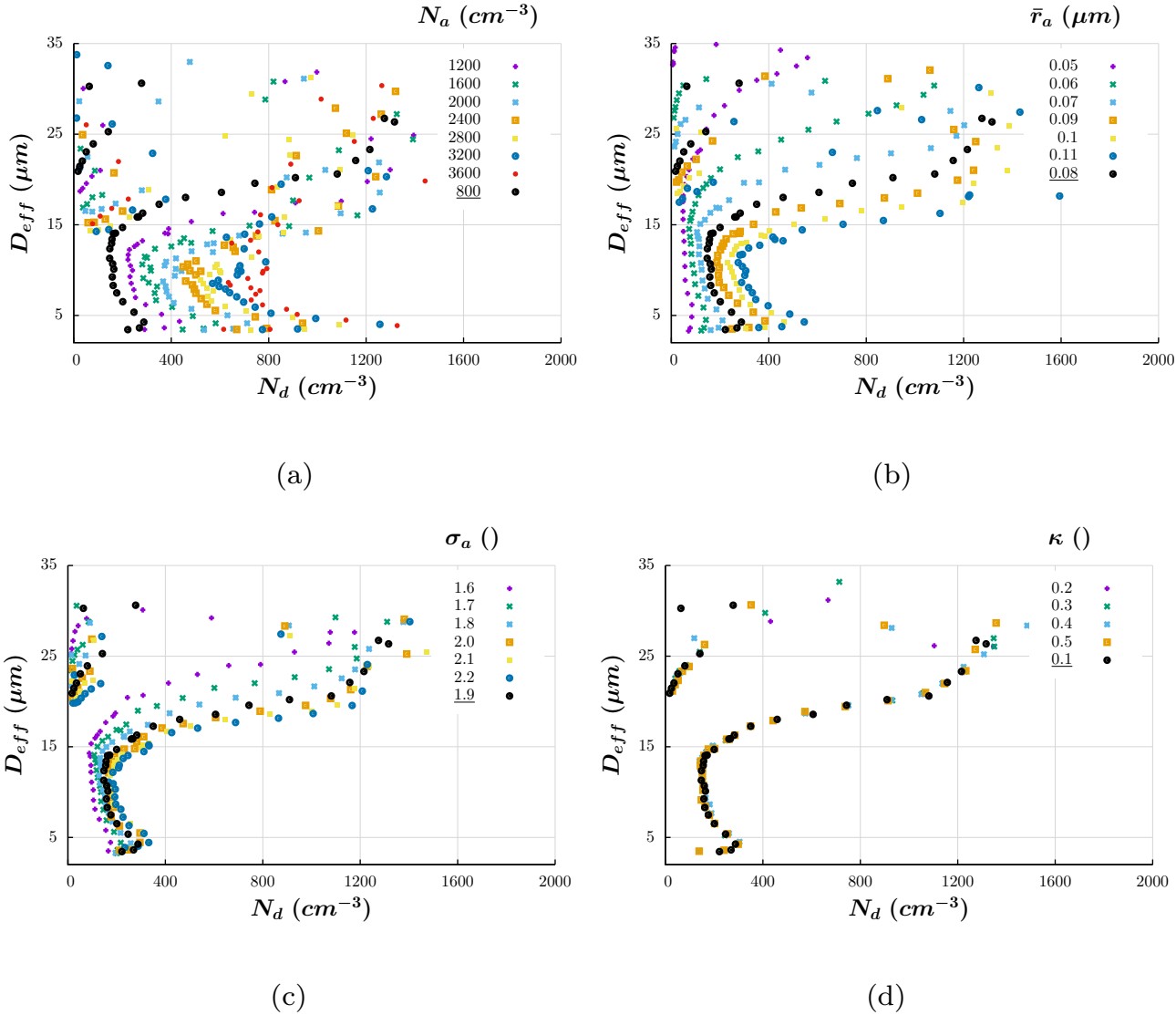

**Figure 3.** Illustration of the sensitivity of cloud-top bulk properties to (a) the aerosol number concentration ($cm^{-3}$), (b) the median radius of the PSD ($\mu$m), (c) the geometric standard deviation of the PSD (), and (d) the aerosol hygroscopicity (). The markers represent the averaged DSDs for the time steps when the cloud top remains at the same model level during its growth. The colors distinguish between simulations using different values of the parameter specified at the top of the graphs. The control simulation is represented by black markers in the figures.

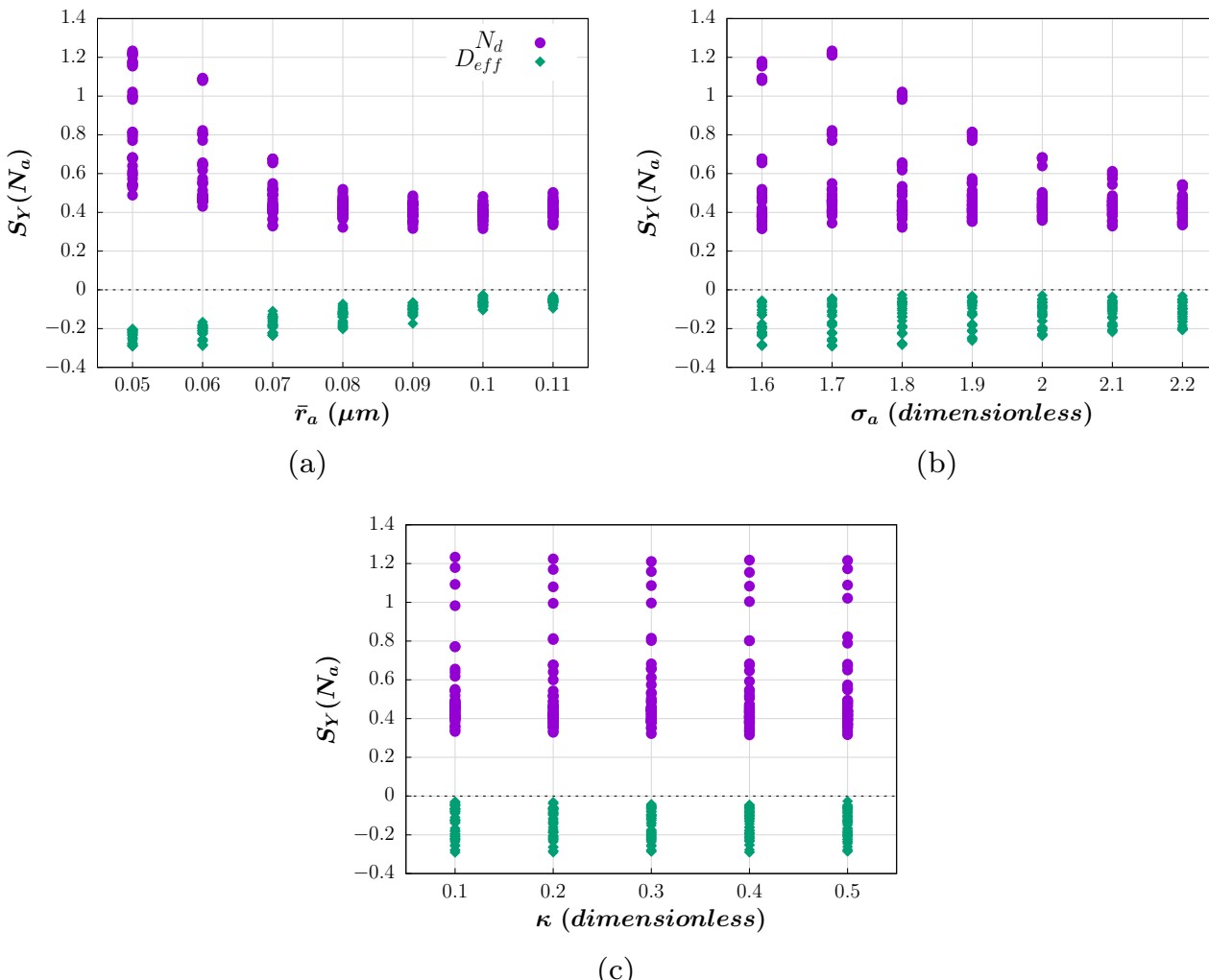

**Figure 4.** Sensitivities of the droplet number concentration and effective diameter to the aerosol number concentration ($S_Y(N_a)$) as a function of (a) the median radius of the PSD ($\mu$m), (b) the geometric standard deviation of the PSD () and (c) the aerosol hygroscopicity ().

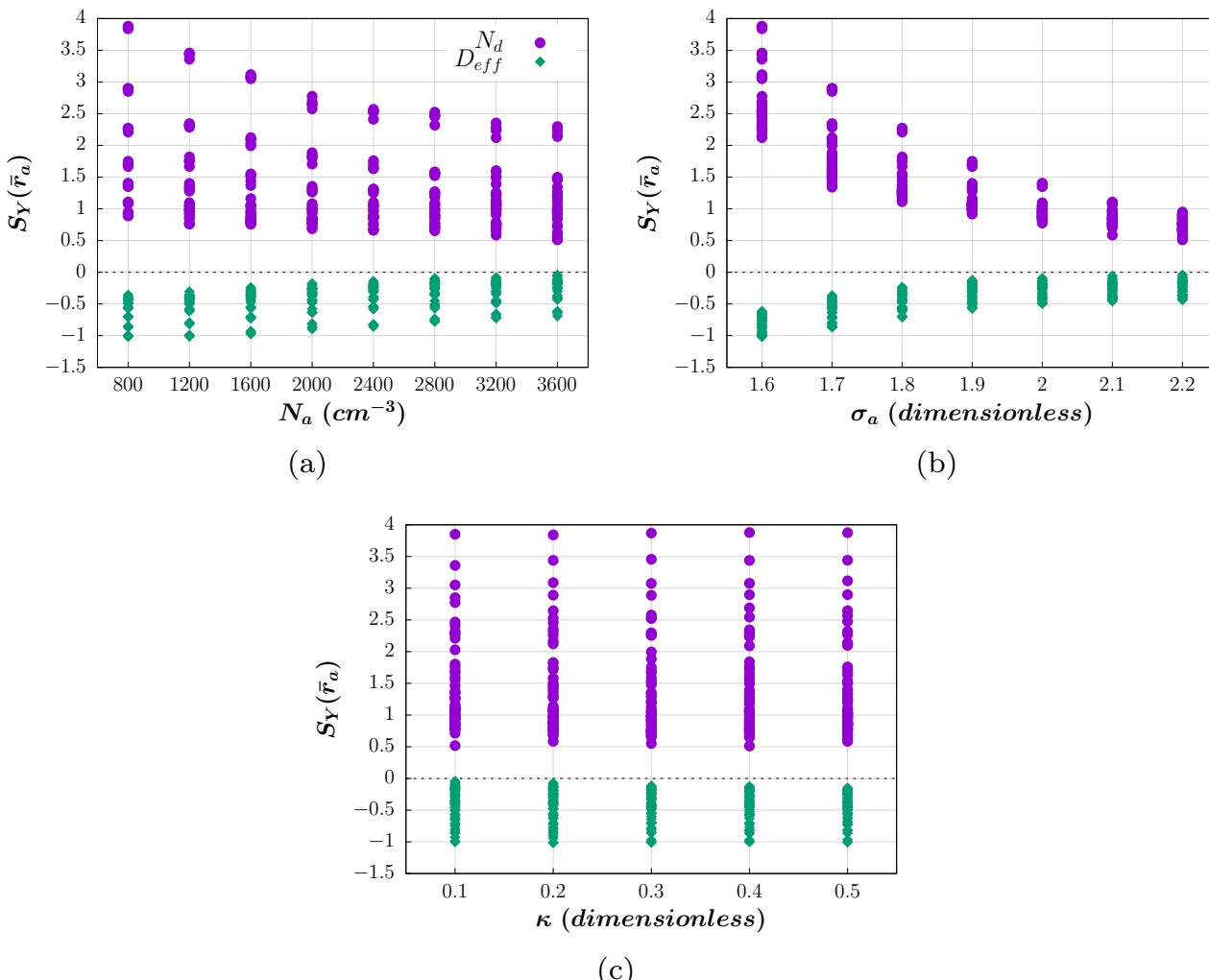

**Figure 5.** Sensitivities of the droplet number concentration and effective diameter to the median radius of the PSD ($S_Y(\bar{r}_a)$) as a function of (a) the aerosol number concentration ($cm^{-3}$), (b) the geometric standard deviation of the PSD () and (c) the aerosol hygroscopicity ().

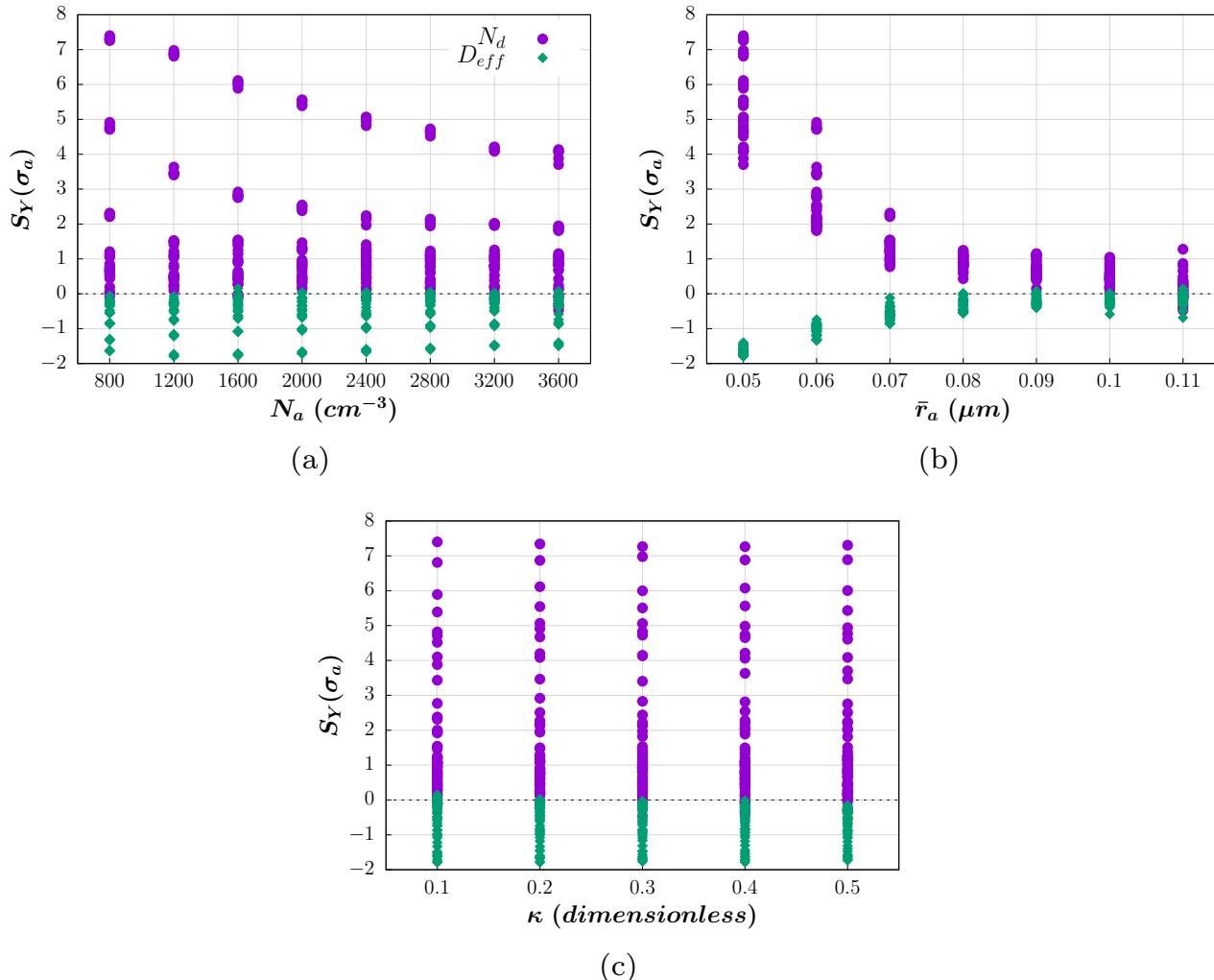

**Figure 6.** Sensitivities of the droplet number concentration and effective diameter to the geometric standard deviation of the PSD ($S_Y(\sigma_a)$) as a function of (a) the aerosol number concentration ($cm^{-3}$), (b) the median radius of the PSD ($\mu$m) and (c) the aerosol hygroscopicity ().

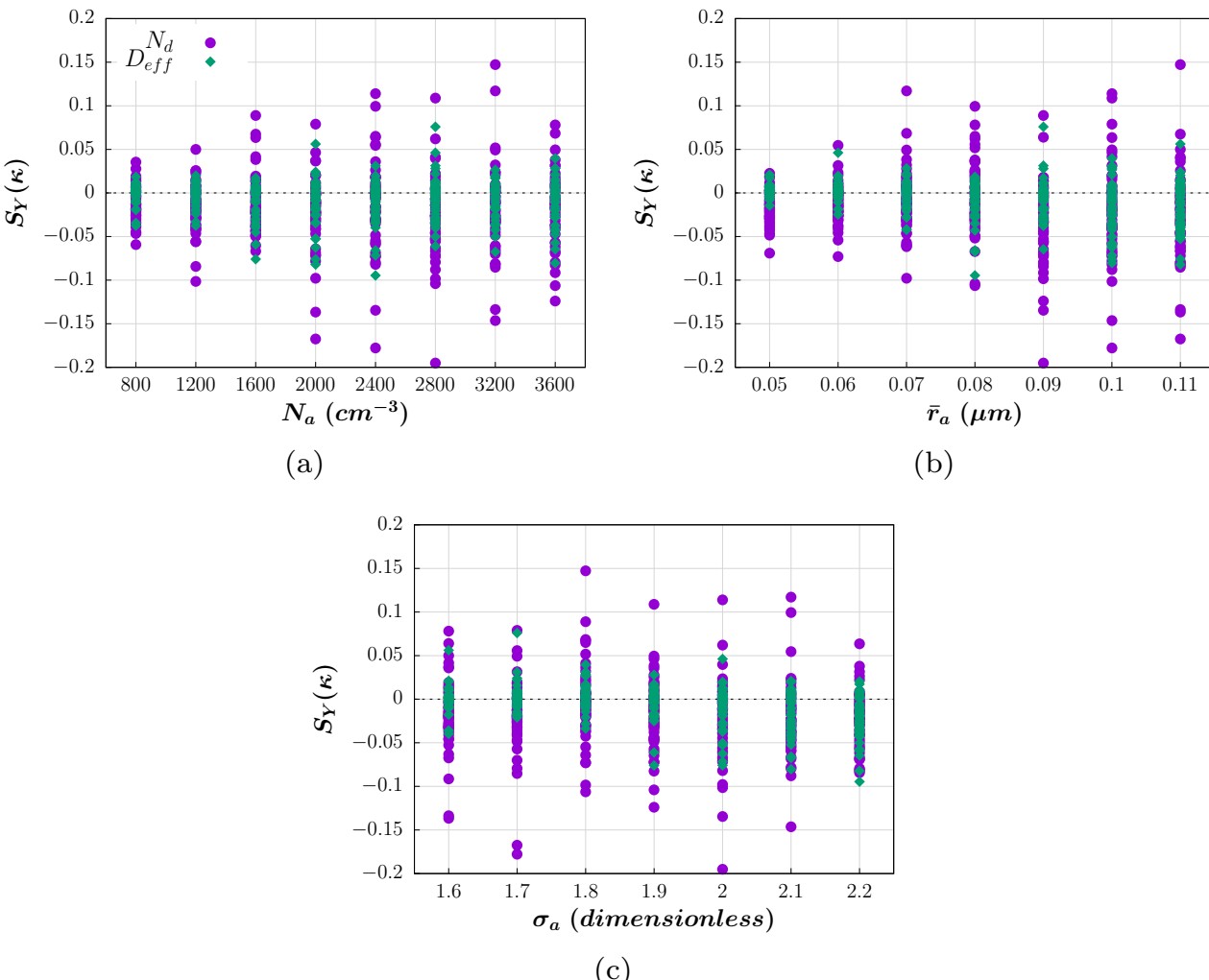

**Figure 7.** Sensitivities of the droplet number concentration and effective diameter to the aerosol hygroscopicity ($S_Y(\kappa)$) as a function of (a) the aerosol number concentration ($cm^{-3}$), (b) the median radius of the PSD ($\mu$m) and (c) the geometric standard deviation of the PSD ().

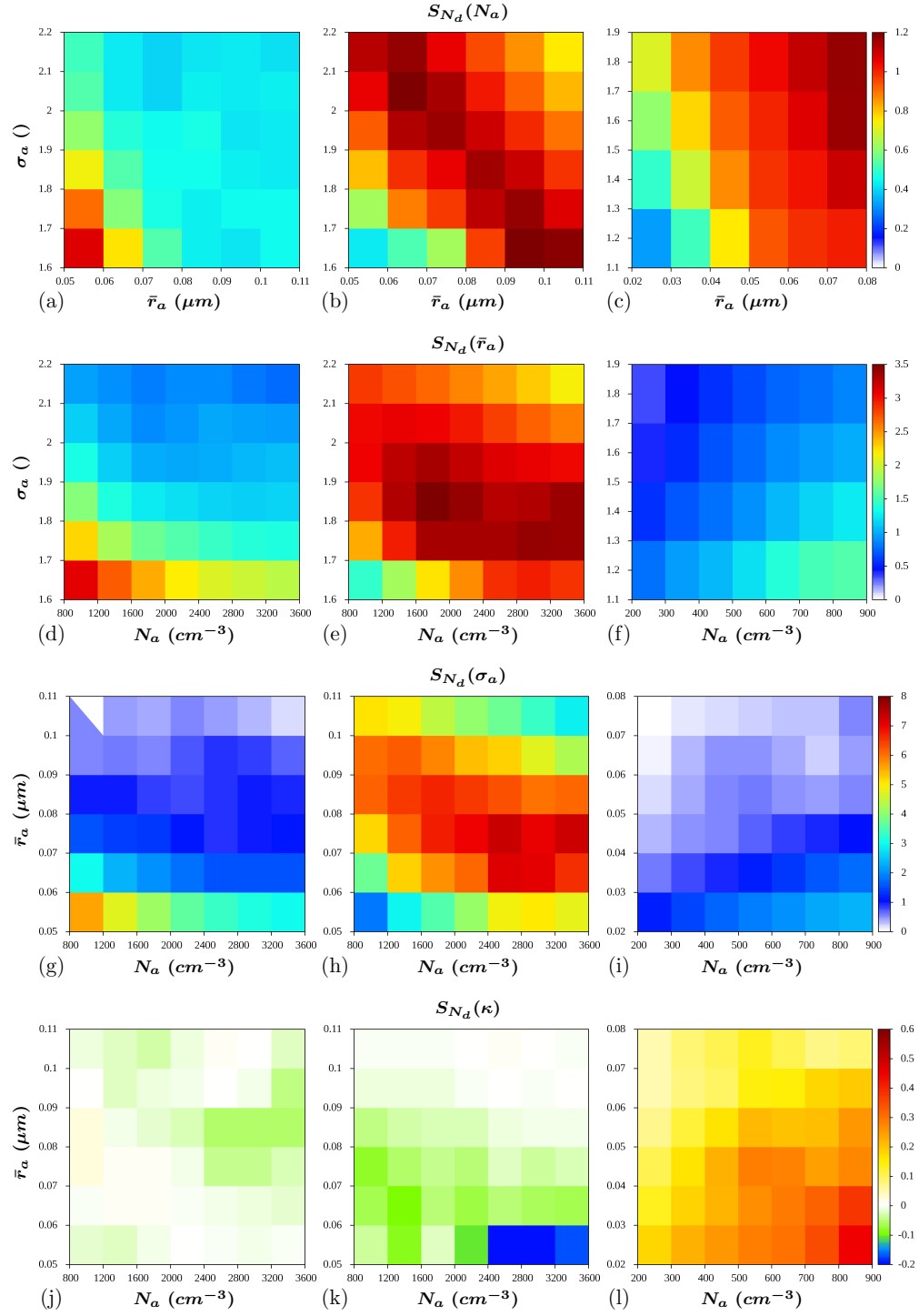

**Figure 8.** Sensitivity of $\bar{N}_d$ to the aerosol properties in three different configurations of the model: with entrainment and bins for the aerosols (a,d,g,j), without entrainment (b,e,h,k) and without bins for the aerosol (c,f,i,l)

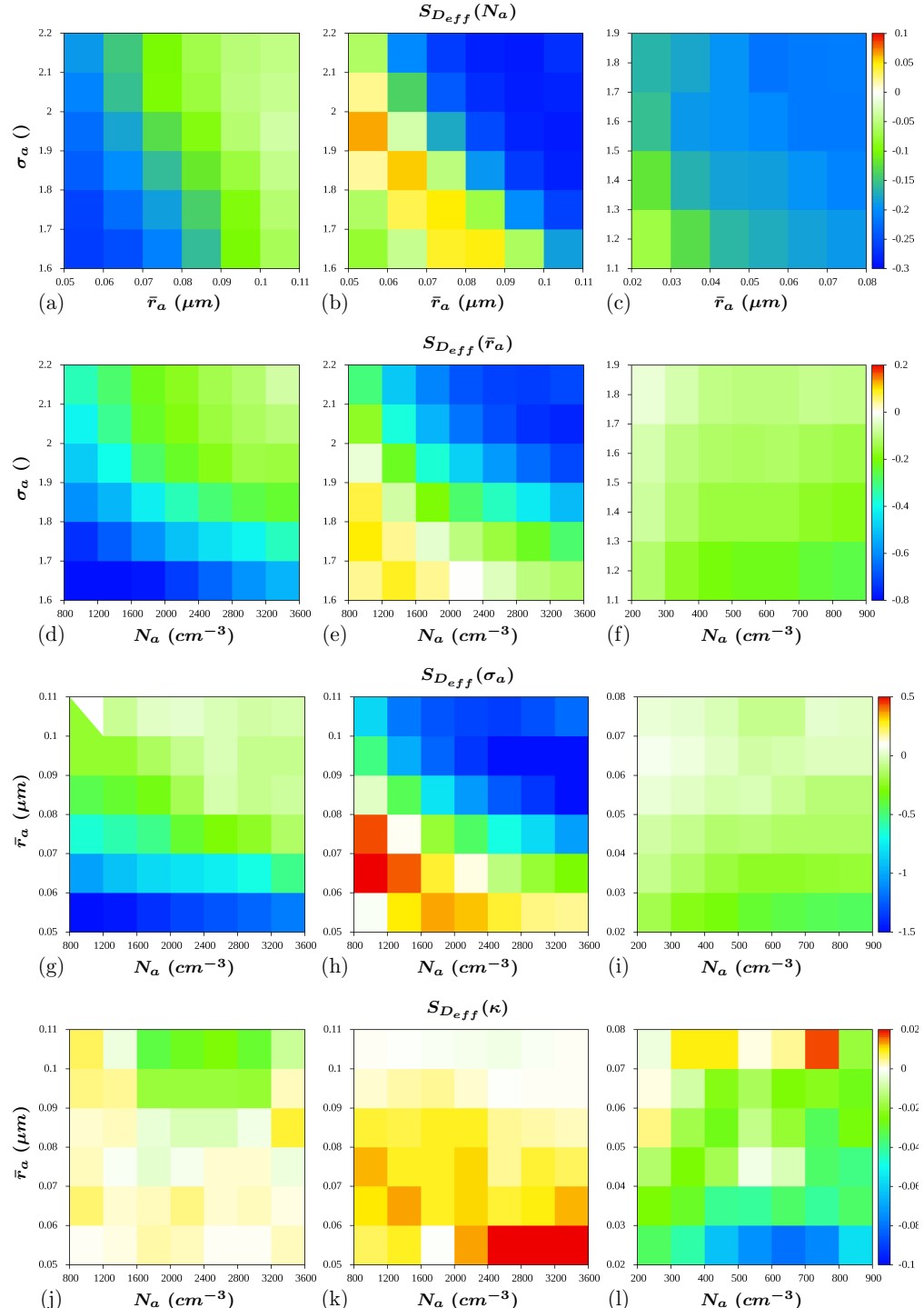

**Figure 9.** Sensitivity of $\bar{D}_{eff}$ to the aerosol properties in three different configurations of the model: with entrainment and bins for the aerosols (a,d,g,j), without entrainment (b,e,h,k) and without bins for the aerosol (c,f,i,l)

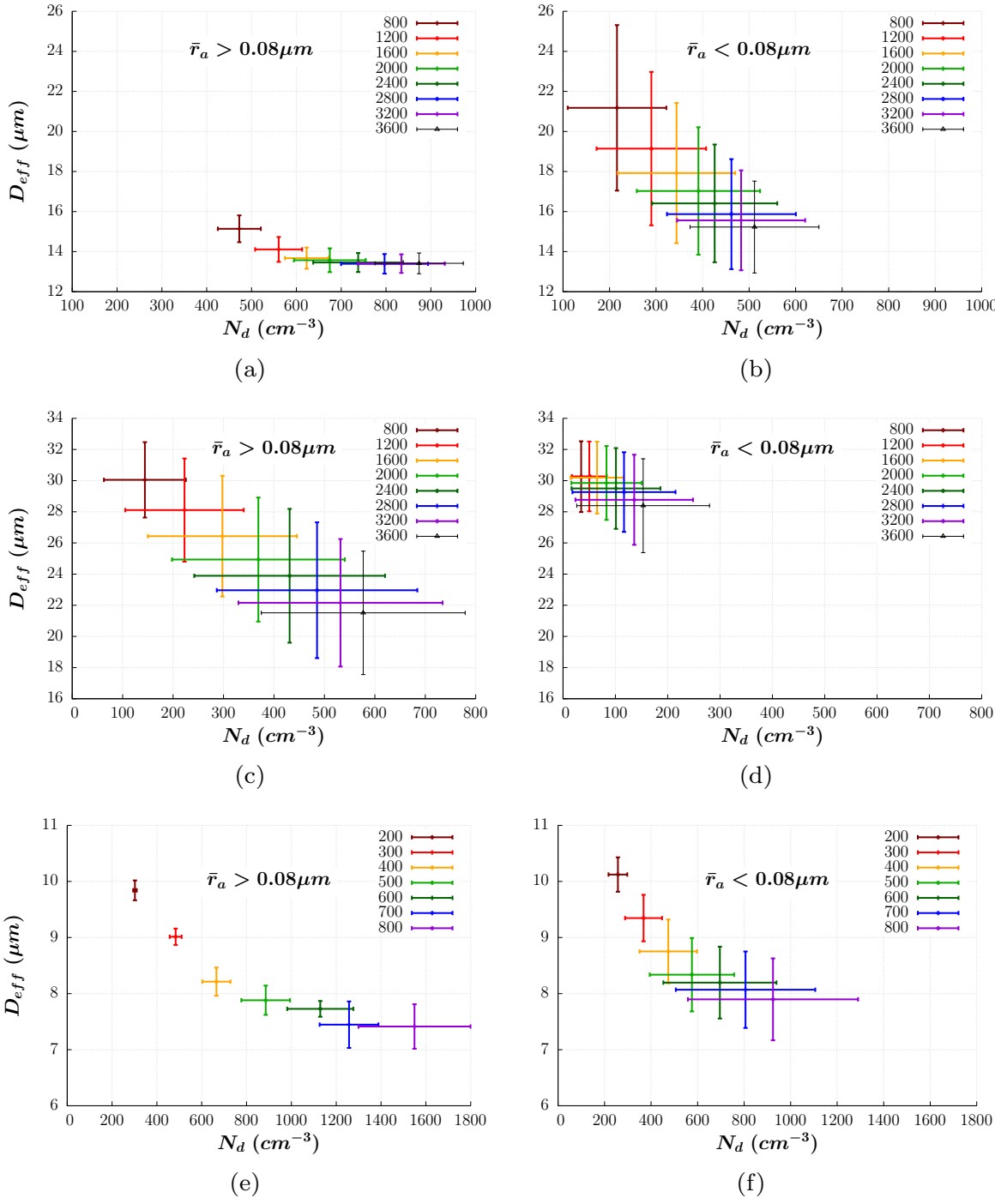

**Figure 10.** Mean and standard deviation of $\bar{N}_d$ and $\bar{D}_{eff}$ at cloud top from the simulations with entrainment and bins for the aerosols (a,b), without entrainment (c,d) and without bins for the aerosol (e,f).