# Peer review of "Quantifying the aerosol effect on droplet size distribution at cloud-top"

_Atmospheric Chemistry and Physics, 2018_

## Referee Comment (RC1) · Anonymous Referee #2 · 2 Jan 2019

Review of acp-2018-1087: "Quantifying the aerosols' effects on droplets size distributions at cloud-top", by LH Pardo et al.

Summary: This is a theoretical study of sensitivities of cloud droplet size distributions to initial aerosol loading. There are two unique aspects in this study: first, the authors limit their discussions on cloud top properties only; second, the sensitivity tests are thoroughly spaced over aerosol characteristics, including total number, median size, standard deviation of a log-normal distribution, and the hygroscopicity. This is a clearly structured manuscript with adequate figures and literature overview. The conclusions agree with various previous studies using different modeling tools and/or with different parameter choices. The main limitation of the current study is the use of a highly simplified kinematic model, albeit with detailed microphysical representations. I understand

that there are tradeoffs to be made in order to carry out a large number of sensitivity tests. However, there should be a much more detailed discussions listing various limitations, and their associated errors, in both the kinematic framework and in handling aerosol activation processes. In addition, I think the scientific quality of the current manuscript can be improved with additional simulations and analyses. I will detail my suggestions in the follow section. There could be significant revisions if the authors decided to carry out some of the additional sensitivity studies.

Main concerns: 1. There are significant limitations in using a kinematic model. In additional, some key aerosol activations processes in the model that have been simplified. The authors skimped some of these limitations here and there in the manuscript. However, they have missed the most important aspect of the limitation discussions, that is, how these simplifications might affect their main conclusions. This is essential if the conclusions were to be useful for understanding aerosol-cloud interactions in the real world. I would suggest that the authors add a discussion section before the conclusion, to carry out some detailed, in-depth discussions. The following is the list of my suggested topics. Some of them are more obvious than others. Some of them are totally missing in the manuscript and need careful considerations. a). Will the conclusions change if a full dynamic model were used? b). If the initial sounding and/or vertical velocity profile changed, will it change the conclusions? c). A small cumulus with cloud top below 6km seem to be the closest real world resemblance of the kinematic model setup. A key piece that is missing is the entrainment of environment air, together with additional aerosols, into such a small cumulus. This is not discussed at all in the manuscript. The entrainment could come from the cloud bottom, side of the cloud, and most challenging, from the cloud top. Since the focus of this study is the cloud top properties, the variations in the cloud top entrainment along might change the existing conclusions. I think that the entrainment can be added fairly easily in the kinematic framework, with pre-determined entrainment rates and vertical variations. I suggest that the authors repeat their calculations with various entrainment rates, repeat the analysis, and see if the conclusions remain the same. I am particularly interested

in how the cloud top properties change if entrainment from the top is added. I believe these additional simulations will improve the scientific quality of this study significantly. d). Prognostic aerosol activation is another significant limitation of the current study. On P4, L24, the authors stated that they use "a 0.25 factor that attempts to accommodate for the fact that not all CCN will grow to the size of the first droplet bin." Please discuss in details how the factor of 0.25 was chosen, how this factor could affect aerosol activation and cloud droplet spectra, and how it will affect the sensitivities. e). Since aerosols are represented prognostically, there is no sink term for them in the microphysical calculations. In reality, aerosols are removed in clouds through both activation and wash out. Please discuss how this simplification will affect the conclusions. f). Aerosol sizes also grow with increasing supersaturation, and consume certain amount of water vapor supply. This is not considered in the model. How important is this process?

2. There are significant vertical variations in simulated cloud properties, as shown in Fig. 2. It will be beneficial to conduct the same sensitivity calculations in Fig. 3 for vertically averaged cloud properties, and compare them with the cloud top properties. The results can also be compared with Cecchini et al (2017).

Minor points: 1. P2, L23: "Must of the previous studies" should be "Most. . ."; 2. P3, L28: "1 s" should be "1s", so is "1200 s"; 3. P8, L5: "Thus the width of the aerosol spectrum can be more important for droplet activation than. . .". I don't agree with this statement. Calculations in Fig. 6 have different units. One cannot compare numbers with different units. 4. Fig. 3: What is the meaning of individual point with the same color? Are they averages over certain time period, or across certain height levels, or something else? 5. It will be nice if the zero lines are labeled in Figs. 4-7, so the positives and negatives can be clearly separated.

---

## Referee Comment (RC2) · Anonymous Referee #1 · 3 Jan 2019

In the manuscript by Pardo et al., the Authors perform a series of simple model based sensitivity tests on aerosol-cloud interactions, with the intention of mapping the sensitivity of cloud properties (number of droplets, droplet size) to several parameters describing the aerosol population. The modelling work is performed with a sectional cloud microphysics scheme coupled to a 1-dimensional column model, which is driven by initial conditions representative of those in the Amazone region and an idealized vertical velocity profile.

Basically, the analysis appears sound, revealing the importance of several aerosol parameters to key cloud microphysical properties. While this is all very interesting, my primary concerns are about the representativeness of the results and the modelling methods used to produce the data for this purpose. Indeed, the Authors state that

the 1-d model (the KiD kinematic driver) is designed mainly for testing microphysical schemes with a consistent kinematic framework. This is true, and in my opinion, it cannot account for important cloud dynamical responses to aerosol perturbations, which we by now know are essential to really understand the aerosol effects on clouds, particularly so in convective cumulus clouds. In particular, I find it rather surprising that the Authors do not consider e.g. how entrainment would affect their results. To back up the representativeness of the results compared to actual clouds, the importance of the dynamics should be somehow evaluated. This would most likely require at least a major review before being published in ACP. I will try to outline my concerns in more detail in the specific comments below.

1. First and foremost, how do you justify using a simple 1-d model, which obviously cannot treat e.g. effects of entrainment, to study aerosol effects on highly dynamic convective cumulus clouds? I agree that you can capture the purely microphysical response with this system (that's what it is designed to do). Even though this is interesting to an extent, I think the results from this setup describe the functionality of the microphysics scheme instead of telling us what we should expect to observe in reality (which can be very different things).

2. The representation of the aerosol size distribution seems very static. I get the impression that cloud activation does not affect the size distribution shape or mean size, just the number. I think this is not a very robust assumption for a study like this. Do the model simulations assume some sort of aerosol replenishment mechanism?

3. Is the assumed vertical velocity profile consistent with the initial temperature profile, if you think about it in terms of releasing an actual thermal? Moreover, since also the evolution of the updraft profile in time is prescribed, do the initial temperature/humidity profiles evolve consistently with the updrafts?

Minor questions:

1. Regarding the discussion on the role of r_a on pages 8-9: the Authors appear to

specify a simple unimodal aerosol size distribution for their simulations (which is fine for this study, I suppose). However, the accumulation mode can most of the time be distinguished in observations and the accumulation mode number concentration specifically is often contrasted to the number of cloud droplets. So doesn't the apparent sensitivity on r_a really fall back to separating the specific mode number concentrations?

2. The manuscript does not really say anything about precipitation. Does precipitation form in the clouds you're simulating? If so, is the aerosol population subject to wet scavenging effects?

––––––––––––––––––––––––––––––––––

---

## Author Response (AR1)

Dear Dr. Ervens,

We appreciate you taking the time to consider our manuscript.

Following the suggestions of both referees, we performed several modifications in the 1D model. In the updated version, instead of prescribing the vertical velocity, it is explicitly calculated from the temperature profile, considering a positive perturbation at surface. The equation for the vertical velocity considers the buoyancy difference between the parcel and the environment, the weight of the condensate and the drag effect of the air in the neighbourhood of the ascending parcel, which includes the effect of the entrainment. The entrainment is parameterized according to the lateral inflow in a vertical jet of radius $R(t,z)$. This process modifies the temperature, humidity and aerosol content inside the parcel. Also, we introduced bins for the aerosol size distribution, and applied a documented methodology to estimate the initial bin for each droplet following activation.

In order to analyze the effect of those approaches, according to the suggestions of the referees, we repeated the simulations for different profiles and physical assumptions. To reflect the new results obtained, the structure of the manuscript was modified. We now present the results of the simulations in three different situations: including a parameterized entrainment and bin for the aerosols, excluding the parameterization of entrainment. and considering the previously employed bulk approach for the aerosols. The first situation is presented as a base case, and a detailed discussion about the corresponding results is addressed in the manuscript. Following, an additional section was introduced to compare the results from the three situation, including new figures.

We believe the manuscript has been benefited from these modifications. In addition to the sensitivity analysis, it now provides a measure of the effects of the physical assumptions considered in the model, in the context of investigating the aerosol-cloud interactions. We hope the modifications that were introduced in the manuscript contributes to provide a better insight into the cloud physics process.

On behalf of all co-authors,

Yours sincerely,

Lianet Hernández Pardo

**(Comment) In the manuscript by Pardo et al., the Authors perform a series of simple model based sensitivity tests on aerosol-cloud interactions, with the intention of mapping the sensitivity of cloud properties (number of droplets, droplet size) to several parameters describing the aerosol population. The modelling work is performed with a sectional cloud microphysics scheme coupled to a 1-dimensional column model, which is driven by initial conditions representative of those in the Amazon region and an idealized vertical velocity profile.**
**Basically, the analysis appears sound, revealing the importance of several aerosol parameters to key cloud microphysical properties. While this is all very interesting, my primary concerns are about the representativeness of the results and the modelling methods used to produce the data for this purpose. Indeed, the Authors state that the 1-d model (the KiD kinematic driver) is designed mainly for testing microphysical schemes with a consistent kinematic framework. This is true, and in my opinion, it cannot account for important cloud dynamical responses to aerosol perturbations, which we by now know are essential to really understand the aerosol effects on clouds, particularly so in convective cumulus clouds. In particular, I find it rather surprising that the Authors do not consider e.g. how entrainment would affect their results. To back up the representativeness of the results compared to actual clouds, the importance of the dynamics should be somehow evaluated. This would most likely require at least a major review before being published in ACP. I will try to outline my concerns in more detail in the specific comments below.**

(Answer) We would like to thank Anonymous Referee #1 for revising our manuscript and suggest improvements. The questions raised were very useful and helped us to consider key aspects in the methods employed and in the analysis of the results. We hope the modifications that were introduced in the manuscript, as a consequence, contributes to provide a better insight into the aerosol-cloud interactions. For practical purposes, we provided a detailed explanation of the modifications implemented in the model, in the introduction of the response to Anonymous Referee #2. In this document we provide responses to the issues raised in the review.

**1. (Comment) First and foremost, how do you justify using a simple 1-d model, which obviously cannot treat e.g. effects of entrainment, to study aerosol effects on highly dynamic convective cumulus clouds? I agree that you can capture the purely microphysical response with this system (that's what it is designed to do). Even though this is interesting to an extent, I think the results from this setup describe the functionality of the microphysics scheme instead of telling us what we should expect to observe in reality (which can be very different things).**

(Answer) We agree the modelling approach employed in this work is highly simplified. In real clouds, there is a much larger variety of process that could enhance or reduce the range of sensitivities that are demonstrated here. Full dynamical models, on the other hand, include many dynamics feedbacks and several subgrid processes that improve the realism of the simulations and provide a more trustable response to aerosol perturbations. However,

performing such a large set of simulations with detailed microphysics and high resolution models is computationally challenging and the interpretation of the results less straightforward. Most of the previous studies using a large subset of simulations have been performed with simple models, such as the adiabatic cloud parcel model of Feingold (2003), Reutter et al. (2009) and Ward et al. (2010).

Although the KiD was designed mainly for testing microphysical schemes with a consistent kinematic framework, without a complete representation of the physical processes other than microphysics, different idealized cases (small cumulus, stratocumulus, deep convection, etc) were elaborated to match observations of real clouds. It is common practice to use idealized cases to understand the responses of the system to different situations, ie., sensitivity tests. The KiD, in particular, was also previously used to analyse physical problems like ice nucleation (Field et al., 2012; Herbert et al., 2015) and aerosol-cloud interaction (Gettelman, 2015).

In our study, we reproduce an idealized cumulus from observed profiles of potential temperature and humidity, using in-situ aircraft observations as a reference. The results of our simulations are found to be consistent with the observations from aircraft penetrations on the top of growing convective clouds over the Amazon, performed in the same day of the sounding used to initialize our model (Cecchini et al., 2017b). Figure RR1 shows the evolution of the cloud-top DSD in the phase-space of the parameters of the gamma function for the observation and the simulation (using the original configuration of the model).

[Figure]

(a)                  (b)

**Figure RR1.** Gamma phase space representation of cloud-top DSDs for different cloud widths: (a) bin microphysics simulation and (b) observation (Fig. 6 of Cecchini et al., 2017b). Small markers represent 1 Hz data, while larger ones are averages for every model level in the simulation and for 200 m vertical intervals in the observation. The color scale represents the height above the cloud base in meters. Projections on axis planes are represented by black continuous lines, in the simulation, and dashed lines, in the observation. The red lines in (a) are the projections of the surfaces with constant $D_{eff}$, increasing from top to bottom.

The differences in absolute values between the graphics from Fig. RR1 are determined by many factors. First, when dealing with the modeled cloud, the boundaries can be quantitatively defined; thus, there is more control over the path that follows the top of the

cloud, as well as the position of the cloud base. Consequently, the initial portion of the graphic that represents the simulation includes information about the very beginning of the cloud, when the first droplets are activated and occupy only one or two bins of the DSD (leading to larger values of μ), while in the graphic that corresponds to the observation, the first DSDs plotted (lower heights above cloud base) correspond to a more developed stage of the cloud. This is why the simulated trajectory looks like an expanded version of the warm portion of the observed one. However, the qualitative similarity between the simulated and observed trajectories is quite remarkable.

The description of the environmental conditions modulates the simulated DSD evolution and is also responsible for similarities and differences between the observed and simulated warm cloud evolution. For example, changes in the initial aerosol concentration can modify the position and shape of the simulated Gamma phase space trajectory by increasing the values of $\Lambda$ and $N_0$ as an expression of more numerous droplets and narrower DSDs. Our sensitivity calculations agree with the calculations of Cecchini et al. (2017a), which use the measurement of the ACRIDICON-CHUVA field campaign at locations with different exposure to anthropogenic aerosol over the Amazon (this comparison is detailed in the response to Anonymous Referee #2).

In order to complement the results already shown in the manuscript, we introduced several modifications in the model, including the treatment of the dynamics and a parameterization of the effect of the entrainment and mixing (a more detailed explanation is provided in the response to Anonymous Referee #2). These new simulations provide an interesting assessment of the effect of increasing the complexity of the process represented by the model. The results obtained from different configurations are consistent, and we believe it would improve our understanding on the importance of those physical processes to evaluate the response of models to different aerosol properties.

**2. (Comment) The representation of the aerosol size distribution seems very static. I get the impression that cloud activation does not affect the size distribution shape or mean size, just the number. I think this is not a very robust assumption for a study like this. Do the model simulations assume some sort of aerosol replenishment mechanism?**

(Answer) Indeed, in the original version of the model, the aerosol size distribution was static, activation and aerosol regeneration did not affect the shape of the PSD, only the total number concentration. To improve the quality of our analysis, we introduced bins for the aerosol and modified the activation and regeneration processes. This had a notable impact on the simulation, mainly due to the CCN depletion. These modifications and its impact on the results are discussed in the response to Anonymous Referee #2.

**3. (Comment) Is the assumed vertical velocity profile consistent with the initial temperature profile, if you think about it in terms of releasing an actual thermal? Moreover, since also the evolution of the updraft profile in time is prescribed, do the initial temperature/humidity profiles evolve consistently with the updrafts?**

(Answer) The vertical velocity profile should depend on the buoyancy force caused by the different densities inside and outside the parcel. Since the vertical velocity was prescribed in

the original version of the model, there was no need for specifying these two different temperatures. The initial temperature profile was considered to be the temperature of the parcel at any time, plus the increase of temperature due to latent heat release. Meanwhile, the water vapor mixing ratio was advected and also modified by the microphysics tendencies.

To improve the consistency between those variables, in the updated version of the model, the vertical velocity is no longer prescribed, instead, it is calculated at any time and height from the instantaneous temperature difference between the parcel and the environment. The atmospheric sounding is used to define the environmental profiles, and a constant temperature perturbation is introduced at surface, in order to cause an upward displacement. The temperature and humidity fields are then advected, as well as the aerosols and the liquid water.

**Minor questions:**

**1. (Comment) Regarding the discussion on the role of r_a on pages 8-9: the Authors appear to specify a simple unimodal aerosol size distribution for their simulations (which is fine for this study, I suppose). However, the accumulation mode can most of the time be distinguished in observations and the accumulation mode number concentration specifically is often contrasted to the number of cloud droplets. So doesn't the apparent sensitivity on r_a really fall back to separating the specific mode number concentrations?**

(Answer) In a sense, yes. However, using $r_a$ is a better approach than simply separating the modes, as it provides a gradual relation between cloud microphysical properties and aerosol size. For parameterization purposes, for instance, it is more desirable to have the $r_a$ and not the mode dependency.

**2. (Comment) The manuscript does not really say anything about precipitation. Does precipitation form in the clouds you're simulating? If so, is the aerosol population subject to wet scavenging effects?**

(Answer) The precipitation is allowed to form in the model, but the amount of rain droplets is very low, specially at cloud top, we therefore neglect the effect of aerosol washout in the simulations.

**References**

CECCHINI, Micael A. et al. Sensitivities of Amazonian clouds to aerosols and updraft speed. Atmospheric Chemistry and Physics, v. 17, n. 16, p. 10037-10050, 2017a.

CECCHINI, Micael A. et al. Illustration of microphysical processes in Amazonian deep convective clouds in the gamma phase space: introduction and potential applications. Atmospheric Chemistry and Physics, v. 17, n. 23, p. 14727-14746, 2017b.

FEINGOLD, Graham. Modeling of the first indirect effect: Analysis of measurement requirements. Geophysical research letters, v. 30, n. 19, 2003.

FIELD, P. R. et al. Ice in clouds experiment–layer clouds. Part II: Testing characteristics of heterogeneous ice formation in lee wave clouds. Journal of the Atmospheric Sciences, v. 69, n. 3, p. 1066-1079, 2012.

GETTELMAN, A. Putting the clouds back in aerosol–cloud interactions. Atmospheric Chemistry and Physics, v. 15, n. 21, p. 12397-12411, 2015.

HERBERT, Ross J. et al. Sensitivity of liquid clouds to homogenous freezing parameterizations. Geophysical research letters, v. 42, n. 5, p. 1599-1605, 2015.

REUTTER, Philipp et al. Aerosol-and updraft-limited regimes of cloud droplet formation: influence of particle number, size and hygroscopicity on the activation of cloud condensation nuclei (CCN). Atmospheric Chemistry and Physics, v. 9, n. 18, p. 7067-7080, 2009.

WARD, D. S. et al. The role of the particle size distribution in assessing aerosol composition effects on simulated droplet activation. Atmospheric Chemistry and Physics, v. 10, n. 12, p. 5435-5447, 2010.

**(Comment) This is a theoretical study of sensitivities of cloud droplet size distributions to initial aerosol loading. There are two unique aspects in this study: first, the authors limit their discussions on cloud top properties only; second, the sensitivity tests are thoroughly spaced over aerosol characteristics, including total number, median size, standard deviation of a log-normal distribution, and the hygroscopicity. This is a clearly structured manuscript with adequate figures and literature overview. The conclusions agree with various previous studies using different modeling tools and/or with different parameter choices. The main limitation of the current study is the use of a highly simplified kinematic model, albeit with detailed microphysical representations. I understand that there are tradeoffs to be made in order to carry out a large number of sensitivity tests. However, there should be a much more detailed discussions listing various limitations, and their associated errors, in both the kinematic framework and in handling aerosol activation processes. In addition, I think the scientific quality of the current manuscript can be improved with additional simulations and analyses. I will detail my suggestions in the follow section. There could be significant revisions if the authors decided to carry out some of the additional sensitivity studies.**

(Answer) We would like to thank Anonymous Referee #2 for taking the time to analyze our work and suggest improvements. We agree that a more detailed discussion on the limitations of the modelling approach, including additional simulations to assess the influence of its shortcomings, would improve the scientific quality of the manuscript. Following the suggestions of both referees, we performed several modifications in the model. The new simulations allowed us to analyse the behavior of the sensitivities in diverse situations, providing a new perspective to the results. We are currently modifying the manuscript hoping to provide a deeper and clearer insight on the results already shown.

In this document we provide detailed responses to the issues raised in the review, as well as a description of the new capabilities of the model.

Major points:

**1. (Comment) There are significant limitations in using a kinematic model. In additional, some key aerosol activations processes in the model that have been simplified. The authors skimped some of these limitations here and there in the manuscript. However, they have missed the most important aspect of the limitation discussions, that is, how these simplifications might affect their main conclusions. This is essential if the conclusions were to be useful for understanding aerosol-cloud interactions in the real world. I would suggest that the authors add a discussion section before the conclusion, to carry out some detailed, in-depth discussions. The following is the list of my suggested topics. Some of them are more obvious than others. Some of them are totally missing in the manuscript and need careful considerations.**

(Answer) In order to address the impact of the limitations in the modelling approach on the results, we have modified two key aspects in in the model: the treatment of the aerosol and the computation of the vertical velocity.

I.    Aerosol:

To better account for changes in the aerosol size distribution, we introduced a set of *19* bins for dry aerosols, with radii (*r*) between *0.0076* and *7.6 µm*, according to Kogan (1991). We consider that the total number concentration of aerosols is log-normally distributed through those bins, at the beginning of the simulation, and can vary by advection, activation and regeneration after droplet evaporation. In-cloud aerosols can also vary by entrainment, which is explained later in this document.

[revised manuscript text omitted]

II.   Vertical velocity

We introduced a new method for estimating the vertical velocity in the model. It is done by solving the simplified vertical momentum equation (Pruppacher and Klett, 1997), considering the buoyancy and the weight of the liquid water, as well as the reaction force on the parcel resulting from the acceleration of the air in the neighborhood (Turner, 1963):

$$\frac{\mathrm{d}W}{\mathrm{d}t} = \frac{g}{1+\gamma}\left(\frac{T-T'}{T'} - \mathrm{w}_L\right) - \frac{\mu}{1+\gamma}W^2$$

where $\gamma \equiv m'/2m \approx 0.5$.

and $\mu$ is the entrainment rate.

For a plume of radius $R_J(z)$, the entrainment rate can be expressed as $\mu_J = C/R_J$, where $C \approx 0.2$ is the entrainment parameter. The equation for the radius of the plume is:

$$\frac{\mathrm{d}\ln R_J}{\mathrm{d}t} = \frac{1}{2}\left[\mu_J W - \frac{\mathrm{d}\ln\rho}{\mathrm{d}t} - \frac{\mathrm{d}\ln W}{\mathrm{d}t}\right]$$

For the case with no entrainment, $\mu_J = 0$ and we also neglect the acceleration of the parcel in the neighborhood, i.e., eliminate the second term and the factor $1/(1+\gamma)$ in the first term in the right side of the vertical velocity equation.

The contributions of the entrainment in the equations for the evolution of the potential temperature, the water vapor mixing ratio and the aerosols is expressed as $\mu_J(X-X')W$, where $X$ and $X'$ represent the in-cloud and environment values for each one of the mentioned magnitudes, respectively.

For the purpose of representing a rising plume, we introduce a constant temperature perturbation at surface. The vertical profile of potential temperature and water vapor mixing ratio are taken from the Boa Vista sounding on 11/09/2014 at 12Z, the same as in the original tests, but no smoothing procedure is applied in this case.

The results obtained with the updated model, with no entrainment, are presented below:

[Figure]

**Figure R1.** Illustration of the sensitivity of cloud-top bulk properties to (a) the aerosol number concentration (cm⁻³), (b) the median radius of the aerosol size distribution (µm), (c) the geometric standard deviation of the aerosol size distribution (dimensionless), and (d) the aerosol hygroscopicity (dimensionless). The markers represent the averaged DSDs for the time steps when the cloud top remains at the same model level during its growth. The colors distinguish between simulations using different values of the parameter specified at the top of the graphs. The control simulation is represented by black markers in the figures.

Figure R1 shows a reduction of the droplet concentration ($N_d$) and an increase of the effective diameter ($D_{eff}$), compared to Fig. 3 in the original manuscript. It is a direct consequence of the modification in the treatment of the aerosol, as explained above. That is the reason why the values of the aerosol parameters are not the same than in the original tests. With the current configuration of the model, when the original values of the parameters are used, there is a very low nucleation rate and the cloud does not develop. It is reasonable, considering that once the aerosol is removed from activation, they are not spread over all sizes as in the previous version of the model, so that no more droplets are nucleated.

The trajectories in Fig. R1 keep the overall shape shown in Fig. 3 (main manuscript) until a critical point, where $N_d$ start decreasing with height. This effect is due to the combination of two factors: the decrease of the nucleation rate and the increase of the collision-coalescence. Note that there is an inverse relation between $N_d$ and $D_{eff}$ at the critical

point in Fig. R1a, i.e., the smaller the aerosol number concentration, the smaller $N_{d,crit}$ and the larger $D_{eff,crit}$. It also happens in the tests with varying $r_a$ and $\sigma_a$, but only for the values larger than the control ones. It evidences that, in the cases with smaller $r_a$ and $\sigma_a$, the decrease of the nucleation rate due to the lack of large aerosols is the dominant factor controlling the upper part of the trajectories. The "saturation" effect that appeared in the original tests is now visible in the tests with varying aerosol number concentration, instead of the tests with the size-related parameters. It indicates the state at which all the supersaturation is consumed, and the system is therefore insensitive to the addition of more aerosols.

[Figure]

**Figure R2.** Sensitivities of the droplet number concentration and effective diameter to the aerosol number concentration ($S_Y(N_a)$) as a function of (a) the median radius of the aerosol size distribution ($\mu$m), (b) the geometric standard deviation of the aerosol size distribution (dimensionless) and (c) the aerosol hygroscopicity (dimensionless).

Figure R2 shows that the sensitivity of $N_d$ to the aerosol number concentration can be almost null for small values of $r_a$ and $\sigma_a$, while having almost no dependency on the aerosol hygroscopicity. It is consistent with the original tests, despite the difference in the values of the parameters tested. However, there is one effect that was not evident in the original tests: a secondary decrease in the sensitivity is found as the aerosol size distribution displaces toward larger aerosols and becomes wider. The latter effect is caused by the supersaturation depletion related to the enhanced activation of aerosols.

The variations in the sensitivity of the droplet effective diameter $D_{eff}$ to the aerosol number concentration $N_a$ are better illustrated in Fig. R3a. It can be observed that it reaches positive values for $\sigma_a = -13.3r_a + 2.7$ approximately, and decreases otherwise. These positive values

are due to absence of water vapour competition. At those points, increasing the aerosol number concentration will create more droplets (note that the sensitivity of Nd to Na is relatively high in that situation), increasing the vertical velocity by latent heat release, and therefore the supersaturation. But the increment in the number of droplets is not as intense as needed to cause a significant water vapour depletion, and since all the droplets will grow in the presence of such high supersaturations, $D_{eff}$ is increased. On the other hand, for the smallest values of $\sigma_a$ and $r_a$, the sensitivity is again negative. In that situation, only the largest aerosols in the right tail of the PSD are activated. Larger drops have a slower rate of growth by condensation, and the collision-coalescence rate may also be decreased due to less variety of fall speeds. Thus, even at high supersaturations, the growth of these droplets can be slower. In addition, when the total number concentration is increased and the shape of the distribution is maintained, the largest increments in the amount of aerosol occur near the center (mode values). Now, let's consider what happens in the right tail of the PSD, i.e., the aerosols that will be activated. In that situation, since the largest increments in number concentration occur toward the center of the distribution, the smaller sizes in the right tail will be favored, leading to a decrease in $D_{eff}$ after activation. If the droplets growth rate is not as intense as to balance that trend, it will result in negative sensitivity.

Overall, Figure R3 shows that increases in both $r_a$ and $N_a$ have a tendency to produce lower $D_{eff}$ (negative sensitivity). However, the effect is controlled by $\sigma_a$. For relatively narrow aerosol PSDs, increases in $N_a$ or $r_a$ have a lesser effect given the limited population of aerosols above the activation diameter. On the other hand, broader aerosol PSDs allow the $r_a$ and $N_a$ effects to go through. In the Amazon, the combination of aerosol sources (e.g. biogenic, biomass burning and urban) can lead to relatively broad aerosol PSDs, suggesting that it is more likely to find negative $D_{eff}$ sensitivities. Cecchini et al. (2017) found an average $S_{Deff}(N_a)$ of -0.25 from aircraft measurements.

[Figure]

**Figure R3.** Sensitivity of the droplet effective diameter to (a) the aerosol number concentration ($S_{Deff}(N_a)$) and (b) the aerosol median radius ($S_{Deff}(r_a)$) as a function of other aerosol properties.

The sensitivity of $N_d$ to the aerosol median radius (Fig. R4) increases for high values of $N_a$ and $\sigma_a$, in agreement with our previous results, but unlike in the original test, has a very small dependency on the aerosol hygroscopicity. Also, the absolute values of $S_{Nd}(r_a)$ in this version

can be more than twice as large as in the original tests. The influence of the depletion of suitable-sized aerosols and water vapor is again visible for the smaller and larger values of $\sigma_a$, respectively, generating a maximum sensitivity at $\sigma_a \approx 1.7$. It reflects also in the behavior of $S_{Deff}(r_a)$, which response to varying Na (Fig. R3a) is similar to the response of $S_{Deff}(N_a)$ to varying $r_a$ (Fig. R3a).

Like in the original tests, the sensitivity to the geometric standard deviation of the aerosol size distribution (Fig. R5) doubles in absolute value and shows a behavior similar to $S_\gamma(r_a)$.

The low values of the sensitivity on the aerosol hygroscopicity (Fig. R6) are consistent with its small influence on the sensitivities of the other parameters, as mentioned above. The trend of its absolute value is similar to the one in the original tests, but the sign of the sensitivities is mostly the opposite. It is reasonable, in this version of the model, because higher values of $\kappa$ define smaller critical radii for activation. Although at first it would increase the droplet number concentration, it also contributes to a faster depletion of the larger aerosols, leading to a reduction in the nucleation rate afterward.

[Figure]

**Figure R4.** Sensitivities of the droplet number concentration and effective diameter to the median radius of the aerosol size distribution ($S_\gamma(\bar{r}_a)$) as a function of (a) the aerosol number concentration ($cm^{-3}$), (b) the geometric standard deviation of the aerosol size distribution (dimensionless) and (c) the aerosol hygroscopicity (dimensionless).

[Figure]

**Figure R5.** Sensitivities of the droplet number concentration and effective diameter to the geometric standard deviation of the aerosol size distribution ($S_Y(\sigma_a)$) as a function of (a) the aerosol number concentration (cm⁻³), (b) the median radius of the aerosol size distribution (μm) and (c) the aerosol hygroscopicity (dimensionless)

[Figure]

**Figure R6.** Sensitivities of the droplet number concentration and effective diameter to the aerosol hygroscopicity ($S_Y(\kappa)$) as a function of (a) the aerosol number concentration (cm⁻³),

(b) the median radius of the aerosol size distribution ($\mu$m) and (c) the geometric standard deviation of the aerosol size distribution (dimensionless).

Finally, aiming to complete the comparison with the original tests, we computed the variability of the cloud droplet bulk properties to emulate the information in Fig. 8 in the main manuscript. Figure R7 shows that the variability of the droplet number concentration and effective diameter (represented by the size of the bars -the standard deviation- in the figure) does not present a significant dependence on the aerosol size, in this case. Instead, the variability is a function of $N_d$ and $D_{eff}$ on their own. In other words, the difference between both graphics resides on the position of the points -for smaller aerosols, $N_d$ will be lower and and $D_{eff}$ will be larger, than for large aerosols-, and that location defines their standard deviation, i.e., points located at the left upper corner in Fig. R7a have approximately the same standard deviation than points at the same location in Fig. R7b.

[Figure]

**Figure R7.** Mean and standard deviation of the time-averaged values of $N_d$ and $D_{eff}$ at the cloud top for each simulation.

**a) (Comment) Will the conclusions change if a full dynamic model were used?**

(Answer) The simulations performed here represent an idealized cloud resulting from observed humidity and temperature profiles and from either a prescribed or prognosed vertical velocity. The control simulation with the original version of the model was previously validated by comparing the evolution of the cloud-top against in-situ observations (see the response to Anonymous Referee #1). Also, the agreement of our results with previous studies regarding the sensitivity to aerosol properties indicates some reliability in our methodology. However, even if we assume it represent a realizable situation, corresponding to an average behavior, it does not include the variety of possibilities existing in real cases. Important processes such as turbulent entrainment and dynamic feedbacks can introduce a significant departure from the idealization we are considering, as Anonymous Referee # 2 pointed out. Full dynamical models account for dynamics feedbacks and several subgrid processes that could enhance or reduce the range of sensitivities that are demonstrated here. Nevertheless, the qualitative behavior of our main results, i.e., the dependency of the cloud sensitivity to the aerosol properties according to its position in the full parameter space, might not change. For example, Gettelman (2015) simulated

several warm rain cases with the KiD and climatological cases with a global model, using a double-moment microphysics scheme, in order to analyze the sensitivity of the aerosol-cloud interaction to cloud microphysics. They found that the test in the KiD were consistent with the global sensitivity tests. This is an aspect we intend to study in a following work, to build on the present results.

**b) (Comment) If the initial sounding and/or vertical velocity profile changed, will it change the conclusions?**

(Answer) In order to address this question, we performed several sets of simulations for increased/decreased values of the potential temperature, the water vapor mixing ratio and the vertical velocity using the original version of the model. The initial profiles of temperature and water vapor were modified by adding/subtracting *0.5*K and *0.5*g/kg, respectively, at all heights. The vertical velocity was modified by means of the maximum updraft speed parameter (*W* in equation 1 in the manuscript) to take values of *4*m/s and *6*m/s. For each one of this modifications, a set of simulations were performed in a way similar to the tests illustrated in Fig. 3 in the manuscript, i.e., when varying one aerosol parameter, the others were fixed at its control values. Then, we calculated the sensitivity $S_{Nd}(X_i)$ and $S_{Deff}(X_i)$ according to equation 2 in the manuscript. The results are summarized in Table 1 and Table 2 below, where the sensitivity for each condition is specified, together with the difference of the sensitivity compared to the control case ("diff" columns) and the percentage this difference represents compared to the control value ("%" columns).

Table 1. Sensitivity of the droplet number concentration to the aerosol parameters specified in the first row.

| | N_a | | | r_a | | | sigma_a | | | kappa | | |
|---|---|---|---|---|---|---|---|---|---|---|---|---|
| | S_Nd | diff | % | S_Nd | diff | % | S_Nd | diff | % | S_Nd | diff | % |
| control | 1,0026 | | | 0,5464 | | | 0,3551 | | | 0,0313 | | |
| q_v-0.5 g/kg | 1,0017 | -0,0009 | 0,09 | 0,5048 | -0,0416 | 7,61 | 0,2600 | -0,0951 | 26,77 | 0,0247 | -0,0066 | 21,05 |
| q_v+0.5 g/kg | 0,9834 | -0,0192 | 1,91 | 0,6180 | 0,0717 | 13,12 | 0,5404 | 0,1853 | 52,18 | 0,0438 | 0,0125 | 39,90 |
| Theta -0.5 K | 0,9911 | -0,0115 | 1,15 | 0,5926 | 0,0462 | 8,46 | 0,4595 | 0,1044 | 29,41 | 0,0395 | 0,0081 | 25,92 |
| Theta +0.5 K | 1,0004 | -0,0022 | 0,22 | 0,5173 | -0,0290 | 5,31 | 0,2836 | -0,0715 | 20,15 | 0,0296 | -0,0017 | 5,54 |
| W=4 | 0,9867 | -0,0158 | 1,58 | 0,5867 | 0,0403 | 7,38 | 0,5423 | 0,1872 | 52,72 | 0,0317 | 0,0003 | 1,01 |
| W=6 | 1,0100 | 0,0075 | 0,75 | 0,5692 | 0,0229 | 4,18 | 0,3144 | -0,0407 | 11,45 | 0,0314 | 0,0000 | 0,11 |

Table 2. Sensitivity of the droplet effective diameter to the aerosol parameters specified in the first row.

| | N_a | | | r_a | | | sigma_a | | | kappa | | |
|---|---|---|---|---|---|---|---|---|---|---|---|---|
| | S_Deff | diff | % | S_Deff | diff | % | S_Deff | diff | % | S_Deff | diff | % |
| control | -0,3522 | | | -0,2245 | | | -0,1433 | | | -0,0125 | | |
| q_v-0.5 g/kg | -0,3611 | 0,0089 | 2,52 | -0,2104 | -0,0140 | 6,25 | -0,0783 | -0,0650 | 45,37 | -0,0177 | 0,0052 | 41,46 |
| q_v+0.5 g/kg | -0,3617 | 0,0095 | 2,69 | -0,2518 | 0,0273 | 12,17 | -0,1891 | 0,0458 | 31,97 | -0,0203 | 0,0078 | 61,95 |
| Theta -0.5 K | -0,3568 | 0,0045 | 1,29 | -0,2441 | 0,0197 | 8,76 | -0,1694 | 0,0261 | 18,19 | -0,0151 | 0,0025 | 20,19 |
| Theta +0.5 K | -0,3590 | 0,0068 | 1,93 | -0,2136 | -0,0108 | 4,82 | -0,0930 | -0,0503 | 35,08 | -0,0137 | 0,0012 | 9,51 |
| W=4 | -0,3411 | -0,0112 | 3,17 | -0,2400 | 0,0156 | 6,93 | -0,2008 | 0,0575 | 40,14 | -0,0157 | 0,0032 | 25,69 |
| W=6 | -0,3617 | 0,0095 | 2,70 | -0,2180 | -0,0065 | 2,89 | -0,1162 | -0,0271 | 18,89 | 0,0545 | 0,0419 | 334,75 |

In Tables 1 and 2, the red (blue) color of cells indicates increased (decreased) sensitivity relative to the control simulation, excepting the sensitivity of the droplet

effective diameter to the higroscopicity parameter ($\kappa$) at *W=6*, to which we will refer later. Yellow cells are for percentage differences higher than *5%*, for reference.

It can be observed in the tables that the sensitivity to the aerosol concentration remains almost unaltered in all cases, with percentage differences relative to the control of less than *3.17%*. The sensitivities to the aerosol median radius and geometric standard deviation result enhanced when the water vapor mixing ratio and temperature are increased (which corresponds to a decreased potential temperature, theta), and reduced otherwise. They are also increased for smaller vertical velocities. For the larger value of the vertical velocity, the sensitivity to these size-related parameters tend to be smaller, despite the fact that $S_{Nd}(r_a)$ is larger for *W=6* than for *W=5* (control value) by a 4.18%. However, its value is still smaller than for *W=4*. Whether that behavior can be related to the existence of a minimum in the sensitivity of $N_d$ for *W=5* is something that needs a much more detailed analysis, with a larger number of simulations for variations in *W*. Nevertheless, given the relatively low value of the percentage difference in both $S_{Nd}$ and $S_{Deff}$, it can also be related to other factors in the calculations, such as those arising from the definition of the cloud-top, for example, or from the error in the fit procedure to calculate *S*. On the other hand, the sensitivities to the aerosol hygroscopicity are very small due to the values of the control parameters, as already commented in the manuscript. The influence of the variations in $q_v$ and theta on the sensitivities to $\kappa$ is the same than for the size-related parameters, except that it seems to exist a minimum of $S_{Deff}$ at the control values of the initial profiles. As mentioned before, a deeper analysis would be necessary to understand its causes. The vertical velocity has almost no influence on $S_{Nd}(\kappa)$, but a relatively huge influence on $S_{Deff}(\kappa)$. The sensitivity of the droplet effective diameter to $\kappa$ is increased by a factor of three at *W=6*, compared to its value at *W=5* (control). Also, besides increasing its modulus, it changes the sign of the sensitivity parameter, meaning that increasing the aerosol hygroscopicity would increase the droplet effective diameter. That behavior is opposite to the expected response, considering that a $\kappa$ is inversely proportional to the critical radius for droplet activation. However, for a given number of aerosol particles, if the updraft is strong enough, the large rate of nucleation can deplete the aerosol content causing the supersaturation to be "used" for increasing droplet sizes thereafter. In that situation, increasing $\kappa$ would accelerate the aerosol depletion, favoring the increase of $D_{eff}$ from then on.

These tests include only a subset of the entire parameter space. To a deeper understanding of the effects of the environmental conditions on the cloud sensitivity to aerosols, it would be necessary to perform a analysis similar to that we present in the manuscript, i.e., simulate all the possible combination of the parameters values over its interval of realizable values.

We are currently performing new tests including variations in the initial profiles, with the updated version of the model. Since the evolution of the variables are better coupled now, it would be interesting to know whether the above results maintain.

**c) (Comment) A small cumulus with cloud top below 6km seem to be the closest real world resemblance of the kinematic model setup. A key piece that is missing is the entrainment of environment air, together with additional**

**aerosols, into such a small cumulus. This is not discussed at all in the manuscript. The entrainment could come from the cloud bottom, side of the cloud, and most challenging, from the cloud top. Since the focus of this study is the cloud top properties, the variations in the cloud top entrainment along might change the existing conclusions. I think that the entrainment can be added fairly easily in the kinematic framework, with pre-determined entrainment rates and vertical variations. I suggest that the authors repeat their calculations with various entrainment rates, repeat the analysis, and see if the conclusions remain the same. I am particularly interested in how the cloud top properties change if entrainment from the top is added. I believe these additional simulations will improve the scientific quality of this study significantly.**

(Answer) As explained above, we introduced a parameterization for the effect of lateral entrainment in the simulations. We consider that the column in the model is located in the center of a plume with radius $R(z)$, which mixes homogeneously with the radially entrained air at each level $z$. The entrainment affects the vertical velocity, the temperature, the humidity and the amount of aerosols in the column. Past studies in the Amazon have assumed that the entrainment mixing in Amazonian clouds is close to the extreme inhomogeneous case, given that the droplet effective radius remain relatively constant horizontally (e.g. Freud et al., 2011). However, the recent studies of Pinsky et al. (2016) and Pinsky and Khain (2018) indicate that homogeneous and inhomogeneous mixing can be indistinguishable for polydisperse DSDs, especially for wide distributions. Additionally, those studies show the inadequacy of previous in-situ techniques to identify mixing type. Based on this finding, we will stick to the homogeneous case in the present study as a first approximation. Further studies would be needed to assess the effects of inhomogeneous mixing and this comparison is beyond the scope of this manuscript.

Some cloud-top mixing is resolved in the model grid. However, it can be affected by the numerical diffusion and dispersion introduced by the scheme that solves the advective terms. In the updated version of the model, we use the Lax-Friedrichs first order, conservative scheme to compute the advection of temperature, water vapor and aerosol, and the ULTIMATE scheme to solve the advection of hydrometeors. A first order upwind scheme is used for solving the vertical velocity equation. The choice of the schemes was done by trial and error, in an attempt to minimize the cloud-top dispersion. However, the representativeness of the mixing induced by such an advection at cloud top must be analyzed carefully, and is out of the scope of this paper. For now, we limit our analysis to the results with and without lateral entrainment, as a proxy for the effect of the dilution caused by mixing with the air in the neighbourhood of the clouds.

Since the entrainment decreases the buoyancy of the rising air, including this process significantly reduces the cloud-top height. In order to obtain a thicker cloud, we increased the temperature perturbation at surface, compared to the no entrainment simulations.

[Figure]

**Figure R8.** Similar to Fig. R1 but for the tests with entrainment.

Figure R8 shows the cloud-top trajectories for several combinations of the aerosol properties in the case including the entrainment. Compared to the no-entrainment cases (Fig. R1), there is an increase of $N_d$ and a decrease of $D_{eff}$, better approaching the behavior reflected in the original tests. The inverse relation for $N_d$ and $D_{eff}$ at the critical point (point where there is a change in the monotonicity of $N_d$) holds for all the combinations of the size-related parameters, which evidences the neutralization of the aerosol depletion effect in this case.

[Figure]

**Figure R9.** Sensitivity of the droplet number concentration to (a, c), the aerosol number concentration ($S_{Nd}(N_a)$) and (b, d) the aerosol median radius ($S_{Nd}(r_a)$) as a function of the other aerosol properties. (a,b) tests without entrainment, (c,d) tests with entrainment.

In Fig. R9, it can be seen that the effect of the entrainment in the sensitivity to the aerosol properties is to change the relative importance of the aerosol and water vapor depletion effects. In the no-entrainment case (Fig. R9a,b), the effect of the aerosol depletion predominates over the effect of the water vapor depletion, for smaller-sized and less numerous aerosols. On the other hand, the effect of the consumption of the supersaturation by large, numerous aerosols, is more evident in the case including entrainment (Fig. R9e,f). It is caused by both the increase of nucleation and the mixing with the entrained, drier air. In that case, even small, sparser aerosols do not cause a significant reduction of the sensitivities, because of the supply of aerosols by entrainment.

[Figure]

**Figure R10.** Similar to Fig. R7 but for the tests with entrainment.

Figure R10 illustrates the variability of the bulk properties of the droplet size distribution for the tests with entrainment. It shows that, in a better agreement with the original tests, the variability of Nd and Deff is considerably larger for the simulations with smaller aerosols.

**d) (Comment) Prognostic aerosol activation is another significant limitation of the current study. On P4, L24, the authors stated that they use "a 0.25 factor that attempts to accommodate for the fact that not all CCN will grow to the size of the first droplet bin." Please discuss in details how the factor of 0.25 was chosen, how this factor could affect aerosol activation and cloud droplet spectra, and how it will affect the sensitivities.**

(Answer) We agree that we were not clear about its meaning, origin and importance. We limited to just mention it, considering that it is a feature of the parameterization (Stevens et al., 1996). We also didn't express its function correctly, in order to do so, it is necessary to clarify that this artifice only applies to the mass increment in the first bin, not to its number concentration.The mass of each bin is a key feature in the TAU scheme, since it employs the method of moments for the calculations of vapor deposition and collection. However, given that the first bin contains very small droplets, the application of this factor does not significantly influence the results. In the new version of the model, the mentioned factor is not considered.

**e) (Comment) Since aerosols are represented prognostically, there is no sink term for them in the microphysical calculations. In reality, aerosols are removed in clouds through both activation and wash out. Please discuss how this simplification will affect the conclusions.**

(Answer) As explained earlier in this document, in the original version of the model, the total number concentration of aerosols was modified by activation, advection, and regeneration although fixing its size distribution. In the updated version, the introduction of bins for the aerosol number concentration allows to represent the evolution of the aerosol size distribution as well, and aerosols are also modified by

entrainment and mixing. Washout is not included, since the amount of precipitation produced in the simulations is negligible.

**f) (Comment) Aerosol sizes also grow with increasing supersaturation, and consume certain amount of water vapor supply. This is not considered in the model. How important is this process?**

(Answer) The consumption of water vapor by pre-activated aerosols is not considered in the model. The only sink of water vapor we consider is the droplet activation. We assume that aerosols smaller than the activation size don't represent a significant sink of water vapor, given the great availability of humidity over the Amazon.

**2. (Comment) There are significant vertical variations in simulated cloud properties, as shown in Fig. 2. It will be beneficial to conduct the same sensitivity calculations in Fig. 3 for vertically averaged cloud properties, and compare them with the cloud top properties. The results can also be compared with Cecchini et al (2017).**

(Answer) Cecchini et al. (2017) used the measurements of aircraft penetrations at the top of growing cumulus to analyse the sensitivity of the droplets population to the aerosol loading, vertical velocity and cloud-top height (taken as a proxy for cloud evolution).

In order to compare our results with Cecchini et al. (2017), we calculated the sensitivity of $N_d$ and $D_{eff}$ to the aerosol number concentration at intervals of 200m above cloud base ($H$). For consistency with their results, we consider the average and standard deviation of the sensitivity values for all the subsets ($H, r_a, \sigma_a, \kappa$):

$S_{Nd}(N_a) = 0.82 \pm 0.55$
$S_{Deff}(N_a) = -0.19 \pm 0.08$

The mean sensitivities are very close to the values reported by Cecchini et. al. (2017), although with higher standard deviations due to the much more detailed nature of the simulations as compared to the aircraft measurements.

Minor points:

**1. (Comment) P2, L23: "Must of the previous studies" should be "Most. . .";**

(Answer) This error was corrected in the manuscript.

**2. (Comment) P3,L28: "1 s" should be "1s", so is "1200 s";**

(Answer) This error was corrected in the manuscript.

**3. P8, L5: "Thus the width of the aerosol spectrum can be more important for droplet activation than. . .". I don't agree with this statement.**

(Answer) The intended meaning of this statement is that, since the sensitivity is higher, a given change in the geometric standard deviation of the PSD would modify the DSD more than a proportional change in the other magnitudes. However, the effect of varying a parameter will be determined by its range of possible values. As the variations in the aerosol median radius, total number concentration and composition can be larger, its impact will be more significant.

**4. (Comment) Calculations in Fig. 6 have different units. One cannot compare numbers with different units.**

(Answer) By definition, the sensitivity is dimensionless. That is the reason why the graphs in Fig. 4,5,6,7 can be compared between each other.

**5. (Comment) Fig. 3: What is the meaning of individual point with the same color? Are they averages over certain time period, or across certain height levels, or something else?**

(Answer) Same color points in Fig. 3 apply for averages of the DSDs according to cloud-top height. The text was modified in the manuscript to clarify the meaning of this graph.

**6. (Comment) It will be nice if the zero lines are labeled in Figs. 4-7, so the positives and negatives can be clearly separated.**

(Answer) These figures will be modified in the corrected manuscript.

The previously mentioned saturation effect can be identified for every spectrum of tests in Fig. 3. There is always an interval of values of the tested parameter in which the system becomes less sensitive. The latter has been discussed before in the literature; for example, can vary as a function of $N_a$ and $\bar{r}_a$. Additionally, it is known that the sensitivity to $\kappa$ increases substantially as $\kappa$ decreases (Petters and Kreidenweis, 2007). However, that effect is more or less evident depending on the values of the other parameters. Hence, to characterize the sensitivity of DSDs to aerosol properties, we should explore the multiparameter space composed by all combinations of discrete values of the parameters from its interval of realizable values.

To illustrate that sensitivity variation, we calculated $S_{\bar{N}_d}(X_i)$ and $S_{\bar{D}_{eff}}(X_i)$, with $X_i$ being $N_a$, $\bar{r}_a$, $\sigma_a$ or $\kappa$. $\bar{N}_d$ and $\bar{D}_{eff}$ are the time averages of $N_d$ and $D_{eff}$ at cloud-top for each simulation, respectively. From Eq. 3, $S_{\bar{N}_d}(N_a)$, for example, is the slope of the linear fit between the values of $\bar{N}_d$ and $N_a$ in logarithmic scale, for a given combination of $\bar{r}_a$, $\sigma_a$ and $\kappa$. The sensitivity to one aerosol parameter can then be calculated a number of times equivalent to all possible combinations of the values of the other parameters in Table 1. From its definition, it follows that positive (negative) values of $S_Y(X_i)$ correspond to increasing (decreasing) $Y$ as $X_i$ increases. Also, $|S_Y(X_i)|=1$ means that a given variation in $X_i$ is accompanied by the same absolute variation in $Y$.

Figures 4, 5, 6 and 7 show $S_Y(X_i)$ as a function of all values of $N_a$, $\bar{r}_a$, $\sigma_a$ and $\kappa$ considered. Generally, $\bar{N}_d$ can be almost three times more sensitive to changes in the aerosol parameters than $\bar{D}_{eff}$, which stems from the mathematical definition of these physical magnitudes. Also, the results for $S_Y(N_a)$ agree with the theoretical limits referred in the literature and all sensitivity calculations include the ranges of previously reported values (Feingold, 2003; Reutter et al., 2009; Ward et al., 2010). For each value in the x-axis of figures 4, 5, 6 and 7, there are several combinations of the other two parameters; as a result, there are several points for each value of the x-axis in the figures.

The impact of $N_a$ on cloud droplets depends on the values of $\bar{r}_a$ , $\sigma_a$ and and $\sigma_a$, but does not vary with $\kappa$, as can be seen in Fig. 4. However, this dependency is stronger for the parameters that define the aerosol size distribution, For smaller values of $\bar{r}_a$ and $\sigma_a$, than for $\kappa$. Note that in Fig. 4c, varying $\kappa$ has a small effect on the distribution of the points, compared to the effects of varying $\bar{r}_a$ and $\sigma_a$ in Figs. 4a and 4b, respectively. The points are more dispersed for smaller $\bar{r}_a$ and $\sigma_a$ and tend to $S_Y(N_a)$ reaches its maximum and presents a large dispersion. On the other hand, it tends to be concentrated around a maximum minimum sensitivity value as $\bar{r}_a$ increases. Generally, for smaller values of $\bar{r}_a$ , $\sigma_a$ and $\kappa$ , $S_Y(N_a)$ can be almost null, i.e. no more or less droplets are being formed, nor its size distribution is being modified, regardless of the quantity of aerosol in the environment. In the vicinity of this state, the activation of droplets is being determined by the characteristics of the aerosol, instead of its number concentration. 
[revised manuscript text omitted]

Lorenz, E. N.: Energy and Numerical Weather Prediction, Tellus, 12, 364–373, https://doi.org/10.3402/tellusa.v12i4.9420, 1960.

Machado, L. A. T., Silva Dias, M. A. F., Morales, C., Fisch, G., Vila, D., Albrecht, R., Goodman, S. J., Calheiros, A. J. P., Biscaro, T., Kummerow, C., Cohen, J., Fitzjarrald, D., Nascimento, E. L., Sakamoto, M. S., Cunningham, C., Chaboureau, J.-P., Petersen, W. A., Adams, D. K., Baldini, L., Angelis, C. F., Sapucci, L. F., Salio, P., Barbosa, H. M. J., Landulfo, E., Souza, R. A. F., Blakeslee, R. J., Bailey, J., Freitas, S., Lima, W. F. A., and Tokay, A.: The Chuva Project: How Does Convection Vary across Brazil?, Bulletin of the American Meteorological Society, 95, 1365–1380, https://doi.org/10.1175/BAMS-D-13-00084.1, 2014.

[revised manuscript text omitted]

25 Wendisch, M., Pöschl, U., Andreae, M. O., Machado, L. A. T., Albrecht, R., Schlager, H., Rosenfeld, D., Martin, S. T., Abdelmonem, A., Afchine, A., Araùjo, A. C., Artaxo, P., Aufmhoff, H., Barbosa, H. M. J., Borrmann, S., Braga, R., Buchholz, B., Cecchini, M. A., Costa, A., Curtius, J., Dollner, M., Dorf, M., Dreiling, V., Ebert, V., Ehrlich, A., Ewald, F., Fisch, G., Fix, A., Frank, F., Fütterer, D., Heckl, C., Heidelberg, F., Hüneke, T., Jäkel, E., Järvinen, E., Jurkat, T., Kanter, S., Kästner, U., Kenntner, M., Kesselmeier, J., Klimach, T., Knecht, M., Kohl, R., Kölling, T., Krämer, M., Krüger, M., Krisna, T. C., Lavric, J. V., Longo, K., Mahnke, C., Manzi, A. O., Mayer, B.,
30 Mertes, S., Minikin, A., Molleker, S., Münch, S., Nillius, B., Pfeilsticker, K., Pöhlker, C., Roiger, A., Rose, D., Rosenow, D., Sauer, D., Schnaiter, M., Schneider, J., Schulz, C., de Souza, R. A. F., Spanu, A., Stock, P., Vila, D., Voigt, C., Walser, A., Walter, D., Weigel, R., Weinzierl, B., Werner, F., Yamasoe, M. A., Ziereis, H., Zinner, T., and Zöger, M.: ACRIDICON–CHUVA Campaign: Studying Tropical Deep Convective Clouds and Precipitation over Amazonia Using the New German Research Aircraft HALO, Bulletin of the American Meteorological Society, 97, 1885–1908, https://doi.org/10.1175/BAMS-D-14-00255.1, 2016.

35 Yin, Y., Levin, Z., Reisin, T., and Tzivion, S.: Seeding Convective Clouds with Hygroscopic Flares: Numerical Simulations Using a Cloud Model with Detailed Microphysics, Journal of Applied Meteorology, 39, 1460–1472, https://doi.org/10.1175/1520-0450(2000)039<1460:SCCWHF>2.0.CO;2, 2000a.

Yin, Y., Levin, Z., Reisin, T. G., and Tzivion, S.: The effects of giant cloud condensation nuclei on the development of precipitation in convective clouds — a numerical study, Atmospheric Research, 53, 91 – 116, https://doi.org/https://doi.org/10.1016/S0169-8095(99)00046-0, 2000b.

YIN, Y., CARSLAW, K. S., and FEINGOLD, G.: Vertical transport and processing of aerosols in a mixed-phase convective cloud and the feedback on cloud development, Quarterly Journal of the Royal Meteorological Society, 131, 221–245, https://doi.org/10.1256/qj.03.186, 2005.

[Figure]

**Figure 1.**  Vertical profiles employed as initial conditions  in the simulations

[Figure]

(a)  (b)  (c)

**Figure 2.** Evolution of  $w$ ( m/s),  $N_d$ ( cm$^{-3}$) and  $D_{eff}$ ( $\mu$ m) in the simulation.

[Figure]

**Figure 3.** Illustration of the sensitivity of cloud-top bulk properties to (a) the aerosol number concentration ($cm^{-3}$), (b) the median radius of the  PSD ($\mu m$ $\mu$ m), (c) the geometric standard deviation of the  PSD (), and (d) the aerosol hygroscopicity (). The markers represent the averaged DSDs for the time steps when the cloud top remains at the same model level during its growth. The colors distinguish between simulations using different values of the parameter specified at the top of the graphs. The control simulation is represented by black markers in the figures.

[Figure]

**Figure 4.** Sensitivities of the droplet number concentration and effective diameter to the aerosol number concentration ($S_Y(N_a)$) as a function of (a) the median radius of the  PSD ($\mu m$ $\mu$ m), (b) the geometric standard deviation of the  PSD () and (c) the aerosol hygroscopicity ().

[Figure]

**Figure 5.** Sensitivities of the droplet number concentration and effective diameter to the median radius of the  PSD ($S_Y(\bar{r}_a)$) as a function of (a) the aerosol number concentration ($cm^{-3}$), (b) the geometric standard deviation of the  PSD () and (c) the aerosol hygroscopicity ().

[Figure]

**Figure 6.** Sensitivities of the droplet number concentration and effective diameter to the geometric standard deviation of the  PSD ($S_Y(\sigma_a)$) as a function of (a) the aerosol number concentration ($cm^{-3}$), (b) the median radius of the  PSD ( $\mu$ m) and (c) the aerosol hygroscopicity ().

[Figure]

**Figure 7.** Sensitivities of the droplet number concentration and effective diameter to the aerosol hygroscopicity ($S_Y(\kappa)$) as a function of (a) the aerosol number concentration ($cm^{-3}$), (b) the median radius of the  PSD ($\mu m$ $\mu$ m) and (c) the geometric standard deviation of the  PSD ().

[Figure]

**Figure 8.**  Sensitivity of $\bar{N}_d$ to the  aerosol properties in three different configurations of  the model: with entrainment and   bins for the  aerosols (a,d,g,j), without entrainment (b,e,h,k) and without bins for  the aerosol (c,f,i,l)

[Figure]

**Figure 9.** Sensitivity of $\bar{D}_{eff}$ to the aerosol properties in three different configurations of the model: with entrainment and bins for the aerosols (a,d,g,j), without entrainment (b,e,h,k) and without bins for the aerosol (c,f,i,l)

[Figure]

**Figure 10.** Mean and standard deviation of $\bar{N}_d$ and $\bar{D}_{eff}$ at cloud top from the simulations with entrainment and bins for the aerosols (a,b), without entrainment (c,d) and without bins for the aerosol (e,f).

---

## Author Response (AR2)

Response to Referee #1

**The revised manuscript by Pardo et al. significantly extends the original model analysis by more careful consideration of the aerosol size distribution, particle depletion, entrainment, etc. This does make a much stronger case to support their conclusions, even though the basic modelling framework is still very simplified.**

We thank Referee #1 for taking the time to analyze the modifications introduced in the manuscript. Below, we present a response to the comments raised in the new report.

**1) The previous review comments are addressed by the revised manuscript, but I'd like to ask the Authors to elaborate a little bit on the selection of the (new) ranges for the aerosol concentrations and other distribution parameters.**
**My point is, that for the new bin treatment, the selected aerosol concentrations are quite high across the board. At the same time, the Authors imply that the lower concentrations used for the less accurate bulk method, which are still in the range of several hundreds per cc, were not enough to produce clouds with the bin method (where activation scavenging can be described more realistically).**
**- Does this happen in combination with the low-end mode mean diameters and STD, or is there some other reason?**
**- Would this suggest that more carefully selected ranges for the distribution parameters (other than the particle number) would alleviate this issue?**
**Please elaborate a bit on the choices you make, as the explanation in Section 3 (e.g. page 6, line 8 onwards) is a bit vague.**

The choice of the intervals of values for the aerosol parameters was made in a way that allowed to explore the largest subset of realizable values of the parameters, while keeping a reasonable computation time. For certain combinations of the size distributions parameters, the PSD can be very narrow, with a very small concentration of aerosols larger than the activation threshold. This configuration, along with a small total number concentration of aerosols, generates clouds with very low water content and unrealistically high supersaturations. To prevent this kind of situations, ranges were chosen as to produce averaged droplet number concentrations at cloud-top >10 cm$^{-3}$, while keeping the largest variety possible for each parameter. In order to test the response to $r_a$=0.05$\mu m$, for example, we needed to use values of $N_a$ larger than 800cm$^{-3}$, and $\sigma_a$ larger than 1.6. We did obtain realistic outputs from simulations with lower $N_a$, such as ~200cm$^{-3}$, but only using wider PSDs with larger median sizes. Since we needed a full discrete parameter space, it explains the choice of the described parameters intervals.

Due to a deficient treatment of the activation scavenging, when a bulk treatment of the aerosol is used, the lower values of the aerosol parameters at which a reasonably dense cloud can be generated are much smaller. By not allowing the PSD to freely evolve, there is a continuous, spurius source of large aerosols that induces unrealistically high values of $N_d$ and can unstabilize the model if some thresholds for $N_a$, $r_a$ and $\sigma_a$ (900cm$^{-3}$, 0.08$\mu m$ and 1.9, respectively) are exceeded. Therefore, in this case, the upper and lower limits for each parameter had to be decreased.

This explanation was included in the manuscript (section 3).

**2) As a more minor note, I feel a bit uneasy with statements like "These results show that the study of the aerosol-cloud interaction must include the parameters describing aerosol properties, such as the size distribution, at least for $\bar{r}_a \leq 0.08$ μm." in the discussion section, as it seems to suggests that studies on aerosol-cloud interactions wouldn't consider the size distribution shape. This generally is not the case especially with contemporary microphysical models. Please consider rewording this accordingly.**

[revised manuscript text omitted]

---

## Author Response (AR3)

**Authors' response**

Dear Co-Editor,

On behalf of my co-authors, I would like to thank you for the time and effort taken to review our manuscript. We corrected the manuscript as indicated in the marked-up version below.

Best regards,

Lianet Hernández Pardo

.

[revised manuscript text omitted]